



# Extensive and anomalous grounding line retreat at Vanderford Glacier, Vincennes Bay, Wilkes Land, East Antarctica.

Hannah J. Picton[1*], Chris R. Stokes[1], Stewart S. R. Jamieson[1], Dana Floricioiu[2], Lukas Krieger[2]

[1]Department of Geography, University of Durham, Durham, DH1 3LE, England
[2]Remote Sensing Technology Institute, German Aerospace Center, Wessling, Germany

*Current address and correspondence to*: School of GeoSciences, University of Edinburgh, EH8 9XP, Scotland (hannah.picton@ed.ac.uk)

**Abstract.** Wilkes Land, East Antarctica, has been losing mass at an accelerating rate over recent decades in response to enhanced oceanic forcing. Overlying the Aurora Subglacial Basin, it has been referred to as the 'weak underbelly' of the East Antarctic Ice Sheet and is drained by several major outlet glaciers. Despite their potential importance, few of these glaciers have been studied in detail. This includes the six outlet glaciers which drain into Vincennes Bay, a region recently discovered to have the warmest intrusions of modified Circumpolar Deep Water (mCDW) ever recorded in East Antarctica. Here, we use remotely sensed optical imagery, differential satellite aperture radar interferometry (DInSAR) and datasets of ice surface velocity, ice surface elevation and grounding line position, to investigate ice dynamics between 1963 and 2022. Decadal trends in frontal position are observed across the Vincennes Bay outlet glaciers, potentially correlated to variations in sea ice production. Ice surface velocities were generally stable between 2000 and 2021, with some fluctuations measured across the grounding line of Bond East Glacier. Changes in ice surface elevation were spatially variable, but a clear and consistent thinning trend was measured at Vanderford Glacier between 2003 and 2020. Enhanced rates of ice thinning were seen across each of the Vanderford, Adams, Anzac, and Underwood Glaciers between 2017 and 2020. Most importantly, our results confirm extensive grounding line retreat at Vanderford Glacier, measured at 18.6 km between 1996 and 2020. Such rapid grounding line retreat (0.8 km yr$^{-1}$) is consistent with the notion that warm mCDW is able to access deep cavities formed below the Vanderford Ice Shelf, driving high rates of basal melting. With a retrograde slope observed inland along the Vanderford Trench, such oceanic forcing may have significant implications for the future stability of Vanderford Glacier.

## 1. Introduction

Mass loss from the Antarctic Ice Sheet (AIS), estimated to hold a sea level equivalent (SLE) of 57.9 ± 0.9 m (Morlighem et al., 2020), has accelerated over recent decades (Schröder et al., 2019). Whilst a range of AIS mass balance estimates have been made (Velicogna et al., 2014; Harig & Simons, 2015; McMillan et al., 2015; Martín-Español et al., 2016; Shepherd et al., 2019; Rignot et al., 2019; Schröder et al., 2019; Smith et al., 2020), a recent collation of 24 independent estimates indicates that the AIS lost a total of 2,720 ± 1,390 Gt of ice between 1992 and 2017 (The IMBIE Team, 2018). This negative



mass balance was dominated by loss from the West Antarctic Ice Sheet (WAIS), averaged at a rate of -94 ± 27 Gt yr$^{-1}$ over the same time period (The IMBIE Team, 2018).

Recent mass loss from the WAIS has largely been concentrated within the Amundsen Sea embayment (Feldmann & Levermann, 2015; Harig & Simons, 2015; McMillan et al., 2015; Gardner et al., 2018; Shepherd et al., 2019), driven by

increased discharge from the fast-flowing Pine Island and Thwaites Glaciers (Mouginot et al., 2014; Medley et al., 2014; Christianson et al., 2016; Yu et al., 2018). The ice flow acceleration (Mouginot et al., 2014; Sutterley et al., 2014), dynamic thinning (Pritchard et al., 2009; Flament & Rémy, 2012) and rapid grounding line retreat (Rignot et al., 2014) of these major outlet glaciers has been attributed to ice-shelf thinning and reduced buttressing, a process forced by the wind-driven intrusion of warm modified Circumpolar Deep Water (mCDW) across the continental shelf to sub-ice shelf cavities (Feldmann &

Levermann, 2015; Scambos et al., 2017; Rignot et al., 2019; Pattyn & Morlighem, 2020). Situated on retrograde bed slopes that are grounded well below sea level, there has been widespread concern that the Pine Island and Thwaites Glaciers could be susceptible to marine ice sheet instability (MISI) (Joughin & Alley, 2011; Parizek et al., 2013; Nias et al., 2016; Lhermitte et al., 2020), whereby irreversible grounding line retreat is triggered (Schoof, 2007). Indeed, Favier et al. (2014), Rignot et al. (2014) and Joughin et al. (2014) argue that such unstable retreat may already be underway.


In contrast, the recent mass balance of the East Antarctic Ice Sheet (EAIS), containing a SLE of 52.2 ± 0.7 m (Morlighem et al., 2020), has typically been estimated to either be in equilibrium (Schröder et al., 2019), or slightly positive (Martín-Español et al., 2016; Gardner et al., 2018; Shepherd et al., 2019; Smith et al., 2020; Stokes et al., 2022). The IMBIE Team (2018) calculated a positive mass balance of +5 ± 46 Gt yr$^{-1}$ between 1992 and 2017, with mass gain predominantly

concentrated in the Dronning Maud Land region (Velicogna et al., 2014; Harig & Simons, 2015; Martín-Español et al., 2016; Gardner et al., 2018). However, Rignot et al (2019) have instead suggested that the EAIS has been a significant contributor to recent sea level rise, with an estimated negative mass balance of -57.0 ± 2 m between 1992 and 2017. Whilst the overall mass balance of the EAIS remains uncertain, numerous studies provide strong evidence of dynamic mass loss across the marine-based Wilkes Land sector (McMillan et al., 2015; Martín-Español et al., 2016; Smith et al., 2020; Stokes et al.,

2022). Wang et al. (2021) have suggested that such mass loss has accelerated rapidly over the past two decades, increasing from -6 ± 22 Gt yr$^{-1}$ between 2003 and 2008 to -51 ± 80 Gt yr$^{-1}$ between 2016 and 2018. In addition, Miles et al. (2016) have observed widespread terminus retreat across the region, with 74% of Wilkes Land outlet glaciers measured to retreat between 2000 and 2012.

Wilkes Land is characterised by a 'warm shelf' regime, whereby weaker easterly winds and an absence of dense water formation facilitates the intrusion of warm CDW onto the continental shelf (Thompson et al., 2018; Stokes et al., 2022). The recent increase in mass loss across Wilkes Land has been linked to the enhanced intrusion of such CDW towards sub-ice shelf cavities, accessed through deep subglacial troughs (Miles et al., 2016; Shen et al., 2018). For example, increased ice



discharge (Rignot et al., 2019) and dynamic thinning (Pritchard et al., 2009; Flament & Rémy, 2012; Li et al., 2016;
Shepherd et al., 2019; Schröder et al., 2019) across the primary outlet of Wilkes Land, Totten Glacier (Figure 1b), has been
attributed to enhanced basal melt rates below the Totten Ice Shelf (Roberts et al., 2018; Pelle et al., 2021). Driven by
increased intrusion of mCDW (Khazender et al., 2013; Gwyther et al., 2014; Spence et al., 2014; Li et al., 2015; Greene et
al., 2017; Rignot et al., 2019), access to the main sub-ice shelf cavity is provided by a deep inland subglacial trough
(Greenbaum et al., 2015; Li et al., 2016). Such troughs are observed across the fjord landscape of the Aurora Subglacial
Basin (ASB) over which Wilkes Land lies (Young et al., 2011; Miles et al., 2016).

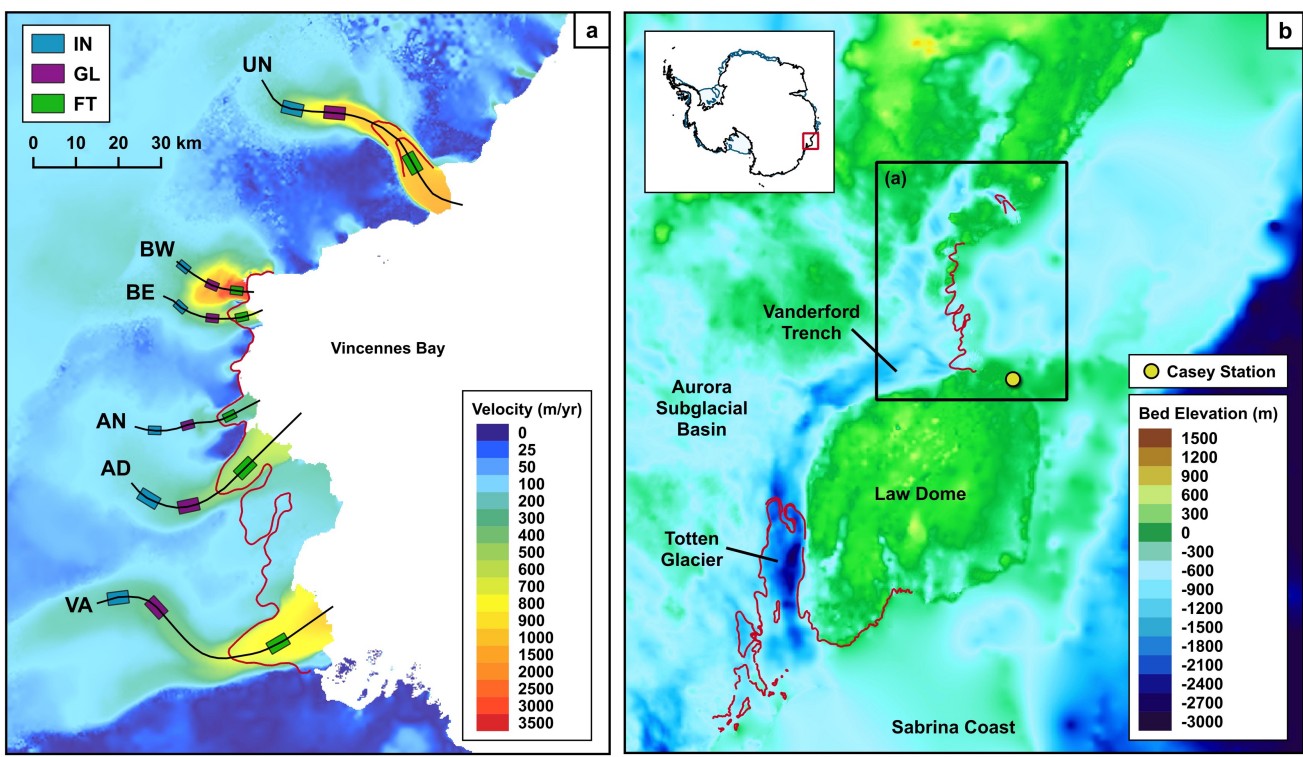

**Figure 1. (a) Ice velocity map of Vincennes Bay extracted from the 2018 ITS_LIVE ice velocity mosaic (Gardner et al., 2022). Central flowlines are digitised down each of the studied glaciers: Vanderford (VA), Adams (AD), Anzac (AN), Bond East (BE),**
**Bond West (BW), and Underwood (UN). Sampling boxes digitised across the inland (IN), grounding line (GL) and floating tongue (FT) areas are shown. (b) Location map of Vincennes Bay and the Aurora Subglacial Basin. Background represents the bed elevation extracted from BedMachine (Morlighem, 2020) and inset map shows the location of the Aurora Subglacial Basin within Antarctica. Red lines in (a) and (b) display the 1996 MEaSUREs grounding line product (Rignot et al., 2016).**

The warmest intrusions of mCDW ever recorded within East Antarctica were observed within Vincennes Bay, defined as the
shelf region situated on the Wilkes Land coast between 104° E and 111° E (Ribeiro et al., 2021). Vincennes Bay overlies the
ASB and is drained by several marine-terminating outlet glaciers: Vanderford, Adams, Anzac, Bond, and Underwood
(Figure 1a). Despite Rignot et al. (2019) briefly noting a dramatic 17 km retreat of Vanderford's grounding line between



1996 and 2017, little attention has been afforded to these outlet glaciers. With relatively high area-averaged basal melt rates
of $5.3 \pm 2$ m yr$^{-1}$ estimated over the Vincennes Bay ice shelves (Rignot et al., 2013), Depoorter et al. (2013) had previously
proposed that Vanderford may be vulnerable to oceanic forcing, potentially driven by the intrusion of mCDW. Indeed, recent
analysis of data collected using instrumented seals revealed ocean temperatures higher than -0.5ºC are able to reach the
Vanderford Ice Shelf, capable of driving basal melt (Ribeiro et al., 2021). Ribeiro et al. (2021) therefore suggested that a
positive feedback may be initiated, whereby continued freshwater input from basal melt could hinder the formation of Dense
Shelf Water (DSW) in the Vanderford polynya, thereby strengthening water column stratification and enabling the enhanced
intrusion of warm mCDW at depth. Analysis by Herraiz-Borreguero & Garabato (2022) also suggests that this potential
feedback may already be underway, with an observed decline in DSW concurrent with consistent sea ice production
indicative of increased penetration of mCDW within the Vincennes Bay region. Whilst Vanderford Glacier is currently
grounded on a prograde bedrock slope (Rignot et al., 2019), it overlies the Vanderford Trench, a deep subglacial trench
characterised by an inland retrograde slope (Figure 1b) (Davis et al., 1986; Chen et al., 2011; Sun et al., 2016). Ice-sheet
modelling indicates that grounding line retreat beyond the present stabilising bedrock ridge could therefore facilitate MISI
(Sun et al., 2016).

This paper seeks to improve our understanding of the largely unstudied Vincennes Bay outlet glaciers, providing an
overview of recent ice dynamics observed between 1963 and 2022. We employ remotely sensed satellite imagery and a
number of secondary datasets in order to analyse variation across four key glacier parameters: (1) terminus position,
manually digitised from satellite imagery; (2) ice surface velocity, extracted from ITS_LIVE (Gardner et al., 2022) and
ENVEO mosaics (ENVEO, 2021); (3) ice surface elevation, assessed using datasets produced by Schröder et al. (2019),
Smith et al. (2020) and Nilsson et al. (2022); and (4) grounding line position, newly mapped using DInSAR techniques and
also extracted from datasets provided by Haran et al. (2005; 2014; 2018), Bindschadler et al. (2011) and Rignot et al. (2016).

## 2. Methods

### 2.1. Image Acquisition

A combination of Landsat 1, 4, 5, 7 and 8, Sentinel-2A and Sentinel-2B images (1973 – 2022) were downloaded from the
USGS EarthExplorer, with the majority of images selected from within the austral summer months of December - February
(Table S1). With each of the Vincennes Bay glaciers occupying a proportionally small area of each scene, cloud cover
thresholds were not applied; instead, images were manually inspected to ensure cloud cover did not obscure the terminus.
Each of the Landsat 1, 4 and 5 scenes were co-registered to a Sentinel-2B scene collected in 2021. Ground control points
(GCPs) digitised across coastal rock outcrops, nunataks, and visibly stable ice features (Glasser et al., 2011) were then used
in order to apply a first-order polynomial transformation with nearest neighbour resampling (Miles, 2013). An orthorectified



mosaic of Antarctica composed of declassified ARGON satellite photographs collected in 1963 was also downloaded (Kim et al., 2007). The spatial resolution of imagery used in this study therefore ranged between 10 and 140 m (Table S1).

## 2.2. Terminus Position

Annual terminus positions were manually digitised within QGIS. It should be noted that Bond Glacier has two distinct outlets separated by an ice rise; these two outlets are referred to as Bond East and Bond West hereafter (Figure 1). In May
2003, failure of the Scan Line Corrector onboard the Landsat 7 satellite resulted in striped data loss (Paul et al., 2017), but the majority of these gaps were observed to be perpendicular to the studied glacier termini. Thus, we were able to digitise terminus positions across the data gaps, with temporally close images with alternative striping patterns used in order to inform each digitisation (Black & Joughin, 2022). Once digitised, changes in terminus position were quantified by applying the well-established box method outlined by Moon & Joughin (2008). This method accounts for asymmetrical changes
across glacier termini, using an open-ended box digitised across the main region of ice flow. Errors associated with this method of analysis arise from the co-registration of satellite imagery, generally estimated at 1 pixel, and manual digitisation of the terminus position, typically calculated as 0.5 pixels (Miles et al., 2018; Miles et al., 2021; Black & Joughin, 2022). Estimated errors associated with each terminus digitisation therefore ranged between 15 and 210 m (Table S1).

## 2.3. Ice Surface Velocity

With a lack of suitable data prior to 2000, the ITS_LIVE (Gardner et al., 2022) and ENVEO (ENVEO, 2021) datasets were used to extract ice surface velocity between 2000 and 2021. ITS_LIVE velocity mosaics are available at an annual resolution between 2000 and 2018 and were derived from Landsat 4, 5, 7 and 8 imagery using auto-RIFT algorithms (Gardner et al., 2018). Each annual velocity mosaic has a spatial resolution of 240 m and reflects the error weighted average of all image-pairs with a centre date that falls within that calendar year (Gardner et al., 2022). In contrast, ENVEO velocity mosaics are
provided at a monthly resolution between 2019 and 2021 and were derived from repeat-pass Sentinel-1 synthetic aperture radar (SAR) data using feature tracking techniques (Nagler et al., 2015). Each of the monthly velocity mosaics has a spatial resolution of 200 m and was processed using the ENVEO software package (ENVEO, 2021). In order to allow for comparison to the ITS_LIVE annual velocity mosaics, mean velocity across each 12-month period was calculated.

Ice surface velocity profiles were extracted along central flowlines manually digitised along each main glacier trunk (Figure 1a). Each longitudinal profile was sampled at an interval spacing of 240 m, reflecting the coarsest spatial resolution of the two velocity datasets used. To analyse changes in ice surface velocity over time, mean annual velocity was extracted from within defined boxes for each given year. In order to assess the spatial variability in ice velocity, three sampling boxes were defined across each glacier; the first box was positioned on the floating tongue (FT), the second located immediately up-
glacier of the most landward observed grounding line position (GL), and the third placed inland a further 5 km upstream (IN) (Figure 1a). Sampling boxes placed over the larger Vanderford, Adams and Underwood Glaciers each covered an area of 15



km², whilst sampling boxes placed across the smaller Anzac, Bond East, and Bond West Glaciers each had a proportionally smaller area of 6 km². Gardner et al. (2022) note that data scarcity is a significant limiting factor in the early ITS_LIVE product, with the auto-RIFT processing chain being limited by the number of image-pairs available across any given year.

Such incomplete coverage was seen across the Vincennes Bay outlet glaciers, particularly prior to the launch of Landsat 8 in February 2013. Average annual velocity values were therefore only extracted from each respective FT, GL and IN box if greater than 25% data coverage was observed (Table S2).

The errors associated with each velocity measurement were also provided at the pixel scale. The mean velocity error was

thus extracted from within each FT, GL and IN box across each given year. The velocity errors provided for each ITS_LIVE velocity mosaic were calculated according to the technical details outlined by Gardner et al. (2022). The errors are updated following co-registration and represent the standard deviation of the difference between the image-pair component velocities and the annual mean component velocities (Gardner et al., 2022). The uncertainty associated with each ENVEO velocity mosaic also represents the standard deviation (ENVEO, 2021). Velocity measurements were not included within the analysis

if the mean error extracted across each box was calculated to be more than 50% of the mean velocity magnitude. However, this threshold resulted in the omission of just 5% of velocity measurements, largely concentrated across Bond West Glacier (Table S2).

**2.4. Ice Surface Elevation**

Variations in ice surface elevation were analysed using the datasets provided by Schröder et al. (2019), Smith et al. (2020)

and Nilsson et al. (2022). Schröder et al. (2019) calculated the monthly surface elevation change (SEC) observed between 1978 and 2017, relative to the reference month 09/2010. SEC measurements were provided at a horizontal resolution of 10 km and were obtained using combined altimetry data from each of the Seasat, Geosat, ERS-1, ERS-2, Envisat, ICESat and CryoSat-2 satellite missions (Schröder et al., 2019). Similarly, Nilsson et al. (2022) calculated the monthly SEC seen between 1985 and 2020, relative to the reference month 12/2013. Produced as part of NASA's ITS_LIVE project, SEC

measurements were provided at a horizontal resolution of 1,920 m and included additional altimetry data from the ICESat-2 mission, launched in October 2018, thus facilitating further coverage between 2018 and 2020 (Nilsson et al., 2022). Mean monthly SEC values provided by both Schröder et al. (2019) and Nilsson et al. (2022) were extracted from each respective grounded GL and IN box. In order to allow comparison between the two datasets, all monthly SEC measurements were calculated relative to 09/1992, representing the earliest shared month for which data coverage was seen across all of the

Vincennes Bay glaciers. Elevation anomalies were then calculated relative to the long-term means derived from each respective dataset between 1992 and 2017.

The ice surface elevation dataset provided by Smith et al. (2020) represents the rate of SEC observed between 2003 and 2019, provided at a horizontal resolution of 5 km using altimetry data from the ICESat and ICESat-2 satellite missions. The





mean rate of SEC observed over this period was extracted from each GL and IN box. In order to allow for comparison across all three elevation datasets, mean rates of SEC were also calculated from the monthly SEC data provided by Schröder et al. (2019) and Nilsson et al. (2022), extracted between 2003 and 2017 and 2003 and 2019, respectively.

**2.5. Grounding Line Position**

Grounding line positions were delineated using differential satellite synthetic aperture radar interferometry (DInSAR) from
ERS-1, ERS-2, and Sentinel-1 imagery collected between 1996 and 2020, processed as part of the European Space Agency's Antarctic Ice Sheet Climate Change Initiative (AIS CCI). The acquisitions used to complete grounding line processing over the Vincennes Bay region are outlined in Table 1. The vertical motion of an ice shelf and hence the hinge line, representing the landward limit of ice flexure, can be observed from a single interferogram (Goldstein et al., 1993). However, the exact location of the hinge line is often obscured by the phase contributions of horizontal ice displacement. Therefore, we typically
selected three repeat pass acquisitions in order to form two interferograms. The difference of such interferograms eliminates the phased contributions from horizontal ice displacement, when assuming a constant ice velocity throughout the observed time period (Rignot, 1996), revealing a dense fringe belt at the grounding zone. The landward limit of this fringe belt was manually delineated as the interferometrically derived grounding line position, with an estimated error of ± 200 m. Within the AIS CCI programme, a processing workflow has been developed to systematically map the grounding line of the AIS
using the Sentinel-1 SAR constellation. However, the repeat orbit of 6 and 12 days depends on the S1 observation plan. Wherever a 6-day repeat from a Sentinal-1 A and B combination was available, these acquisitions were favoured in order to limit temporal decorrelation due to changing surface conditions.

| Satellite | Relative orbit/ Pass Direction | T1 | T2 | T3 |
|-----------|-------------------------------|-----------|-----------|-----------|
| ERS | 231/D | 1996-04-10 | 1996-04-11 | - |
| Sentinel-1 | 133/D | 2016-06-29 | 2016-07-11 | 2016-07-23 |
| Sentinel-1 | 99/A | 2017-07-22 | 2017-07-28 | 2017-08-03 |
| Sentinel-1 | 99/A | 2017-12-01 | 2017-12-07 | 2017-12-13 |
| Sentinel-1 | 99/A | 2017-12-07 | 2017-12-13 | 2017-12-19 |
| Sentinel-1 | 70A | 2020-11-13 | 2020-11-19 | 2020-11-25 |
| Sentinel-1 | 70A | 2020-11-19 | 2020-11-25 | 2020-12-01 |

**Table 1. ERS and Sentinel-1 acquisitions used for interferogram generation in the Vincennes Bay region. Relative orbits and pass direction are stated, with A and D representing ascending and descending pass directions, respectively. T2 was chosen as the primary scene, with two interferograms T2-T1 and T2-T3 then formed. In the case of the ERS satellite, the grounding line was delineated directly on a 1-day repeat pass interferogram.**



A number of secondary grounding line datasets were also employed to enable comparison between datasets and to generate a
higher temporal resolution of grounding line positions (Table 2). The Making Earth Science Data Records for Use in
Research Environments (MEaSUREs) grounding line product was provided using similar DInSAR techniques as previously
described for the AIS CCI product, also applied to ERS-1 and ERS-2 imagery collected in 1996 (Rignot et al., 2016). Whilst
localised variations in positional accuracy are observed (Rignot et al., 2011), the MEaSUREs grounding line product had an
associated standard error of ± 100 m (Rignot et al., 2016) (Table 2).


| Dataset | Date | Method | Error (m) |
|---|---|---|---|
| AIS CCI | 1996 2016 2017 2020 | Grounding line derived using differential satellite synthetic aperture radar interferometry from ERS-1, ERS-2, and Sentinel-1 imagery. | ± 200 |
| MEaSUREs (Rignot et al., 2016). | 1996 | Grounding line derived using differential satellite synthetic aperture radar interferometry from ERS-1 and ERS-2 imagery. | ± 100 |
| ASAID (Bindschadler et al., 2011) | 2001 (1999 – 2003) | Manually digitisation of break-in-slope using observable change in image brightness from Landsat 7 imagery, aided by ICESat surface elevation profiles. | ± 502 |
| MOA (Haran et al., 2005; 2014; 2018) | 2004 2009 2014 | Manually digitised break-in-slope from contrast-enhanced imagery derived from the 2004, 2009 and 2014 Mosaics of Antarctica. | ± 250 |

**Table 2. Details of the different mapping methods employed to derive each of the grounding line datasets used in this study. Note
that the MEaSUREs, ASAID and MOA products represent secondary datasets.**

The Antarctic Surface Accumulation and Ice Discharge (ASAID) grounding line dataset was provided through manual
delineation of the most seaward break-in-slope, observed using a combination of Landsat-7 images collected between 1999
and 2003, and surface elevation data obtained from the ICESat satellite mission (Bindschadler et al., 2011). At outlet glacier
boundaries such as those studied within Vincennes Bay, the ASAID grounding line dataset had an estimated positional error
of ± 502 m (Bindschadler et al., 2011). The Mosaic of Antarctica (MOA) grounding lines were also provided through the
manual digitisation of the most seaward break-in-slope, observed from the 2004 (Haran et al., 2005), 2009 (Haran et al.,
2014), and 2014 (Haran et al., 2018) mosaics, respectively, each composed using MODIS imagery (Scambos et al., 2007).
Each MOA grounding line delineation therefore had an estimated associated error of ± 250 m (Haran et al., 2005; 2014;
2018). Whilst the AIS CCI and MEaSUREs products both represent the inner limit of tidal flexure and thus approximate the
actual grounding line position (Fricker et al., 2009; Rignot et al., 2016), the ASAID and MOA datasets instead reflect the



break-in-slope. This narrow region is observed seaward of the grounding line and is typically inferred to be the surface expression of the abrupt change in basal ice interface produced at the transition between grounded and floating ice (Scambos et al., 2007; Fricker et al., 2009; Bindschadler et al., 2011).

In order to quantify changes in grounding line position, change was measured from the earliest 1996 delineation along each

central flowline (Figure 1a). It should be highlighted that grounding line positions at Vanderford Glacier in 1996 were provided by both AIS CCI and the MEaSUREs datasets, but were digitised in slightly different locations, approximately 640 m apart. Similarly, the MEaSUREs dataset also delineated two different 1996 grounding line positions at Underwood Glacier, separated more significantly by ~ 7 km (Figure 1a). Conservative grounding line retreat values were thus calculated for these glaciers, with retreat measured relative to the most landward observed 1996 position.


**2.6. Bed and Ice Surface Topography**

In order to assess the potential vulnerability to MISI, bedrock elevation profiles were derived from BedMachine (Morlighem, 2020) along each digitised flowline (Figure 1a). Bed elevation values and their associated errors were sampled at an interval spacing of 500 m, reflecting the horizontal resolution of the dataset. Whilst BedMachine was mostly mapped using mass

conservation methods across regions of fast-moving ice, a range of different methods were employed across Vincennes Bay (Figure S1); the sampled errors therefore ranged between 10 and 202 m. Surface topography profiles obtained using the Reference Elevation Model of Antarctica (REMA) (Howat et al., 2019) were also extracted along each central flowline, sampled at the same 500 m interval spacing as the bed elevation.

**3. Results**

**3.1. Terminus Position**

The terminus positions of the Vanderford, Adams, and Underwood Glaciers displayed the greatest variability across the Vincennes Bay outlet glaciers, fluctuating by ~7 km over the study period (Figure 2). Vanderford Glacier retreated ~4 km between 2007 and 2010, with a large extension of ice seen to protrude from the central terminus region (Figure 3a) being removed via calving. Following initial terminus advance, Adams Glacier showed a significant overall retreat, measured at

~6.3 km between 1973 and 2022 (Figure 2b). A spectacular disintegration of Underwood's floating tongue resulted in ~4.6 km of retreat between 2020 and 2022 (Figure 2f), with calved ice blocks clearly visible within the frontal sea ice mélange (Figure 3f). The Bond East and Bond West termini were comparatively stable, fluctuating by a maximum of ~ 1.5 km over the study period (Figure 2). In clear contrast, Anzac Glacier exhibited an overall advance, measured at ~ 2 km between 1963 and 2022 (Figure 2c).






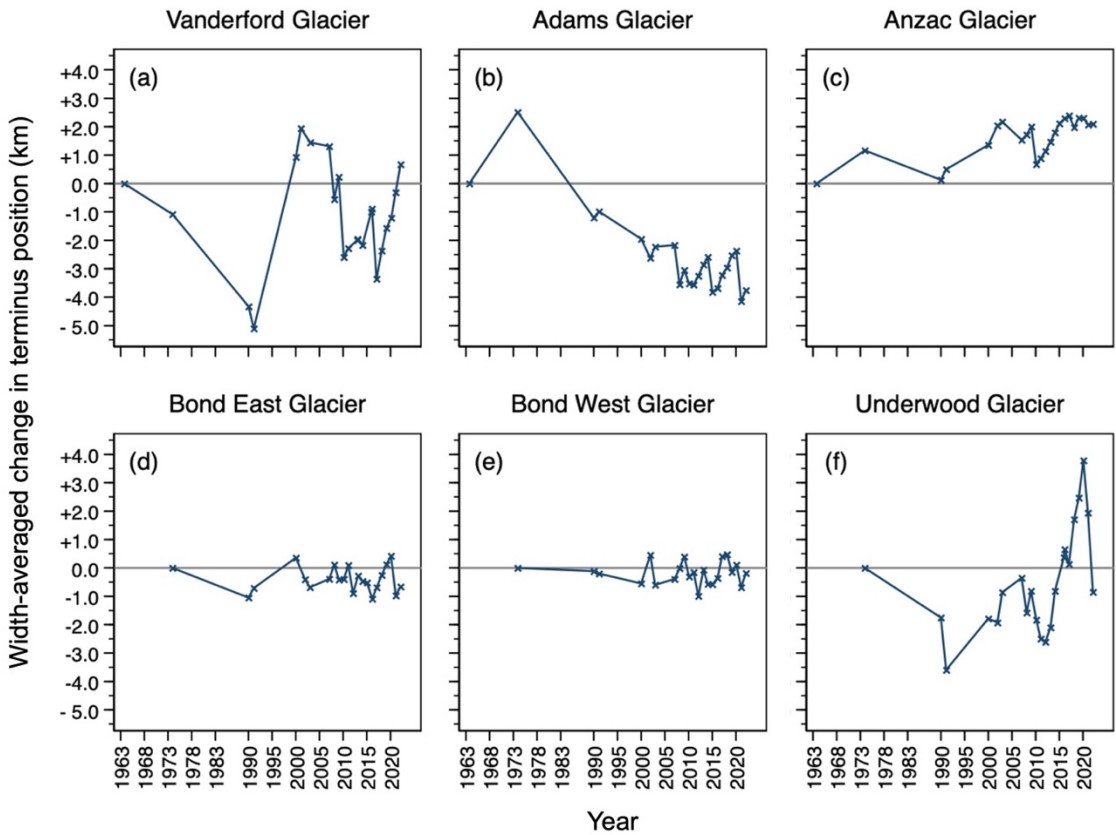

**Figure 2. Width-averaged terminus position change observed relative to the first measurement point at a) Vanderford Glacier, b) Adams Glacier, c) Anzac Glacier, d) Bond East Glacier, e) Bond West Glacier, and f) Underwood Glacier.**



**Figure 3. Minimum (red), maximum (blue), and notably asymmetrical (orange) terminus positions manually digitised across (a) Vanderford Glacier, (b) Adams Glacier, (c) Anzac Glacier, (d) Bond East Glacier, (e) Bond West Glacier, and (f) Underwood Glacier. All background satellite imagery displayed was collected in February 2022, downloaded from the USGS EarthExplorer.**






At the decadal scale, the Vincennes Bay outlet glaciers fluctuated between periods of largely synchronous terminus retreat
and terminus advance. Between 1973 and 1991, each of the six glaciers showed an overall retreat, calculated at a median rate
of -116.8 m yr$^{-1}$ (Table 3). In contrast, between 1991 and 2000, four glaciers were observed to advance, with a positive
median rate of terminus position change of +106.8 m yr$^{-1}$ calculated across the six glaciers. Between 2000 and 2012, all six
glaciers showed an overall retreat, calculated at a median rate of -86.3 m yr$^{-1}$. Between 2012 and 2022, five glaciers were
recorded to advance, with a median rate of +88.1 m yr$^{-1}$ thus observed across the Vincennes Bay outlet glaciers.


### Rate of Terminus Position Change (m yr$^{-1}$)

|  | 1973 - 1991 | 1991 - 2000 | 2000 - 2012 | 2012 - 2022 |
| --- | --- | --- | --- | --- |
| **Mean** | -117.4 | +156.5 | -96.6 | +98.6 |
| **Median** | -116.8 | +106.8 | -86.3 | +88.1 |
| **STD** | 97.8 | 275.3 | 80.5 | 111.7 |

**Table 3. Rate of terminus position change observed across the Vincennes Bay outlet glaciers.**

### 3.2. Ice Surface Velocity

Significant variability in ice surface velocity was observed between the Vincennes Bay outlet glaciers (Figure 1a), with
spatial variations also seen along-flow at each glacier (Figure 4). In 2018, Anzac Glacier was measured to be the slowest
flowing outlet glacier, with velocity increasing from $128 \pm 1$ m yr$^{-1}$ inland at AN (Figure 1a) to a maximum of $331 \pm 1$ m yr$^{-1}$
at the terminus (Figure 4c). In contrast, Bond West was measured to be the fastest flowing outlet glacier, accelerating rapidly
from $241 \pm 4$ m yr$^{-1}$ inland at BW (Figure 1a) to a maximum velocity of $3,339 \pm 2$ m yr$^{-1}$ (Figure 4e), representing one of the
highest recorded velocities within Antarctica. The maximum velocity measured across each of the Vanderford, Adams, Bond
East and Underwood Glaciers ranged from $598 \pm 1$ to $1,625 \pm 4$ m yr$^{-1}$ (Figure 4).

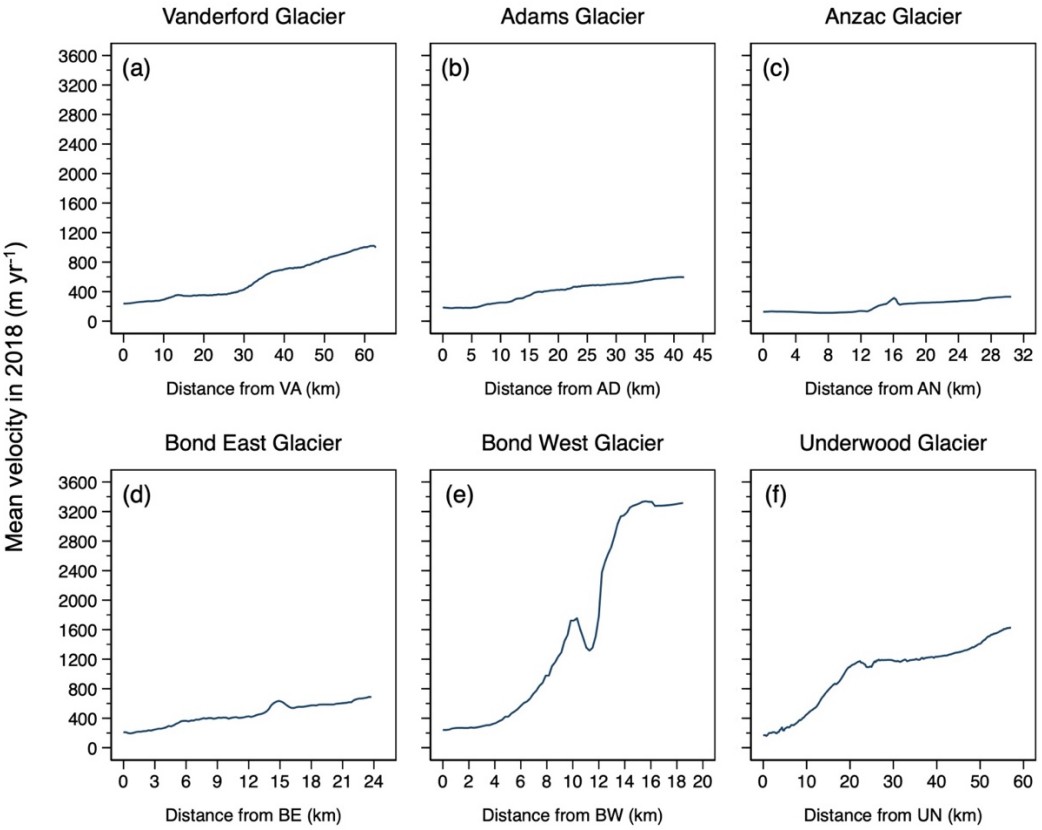

**Figure 4. Ice surface velocity profiles extracted using the ITS_LIVE 2018 velocity mosaic (Gardner et al., 2022) along the central flowlines of (a) Vanderford Glacier, (b) Adams Glacier, (c) Anzac Glacier, (d) Bond East Glacier, (e) Bond West Glacier, and (f)**
**Underwood Glacier.**

Temporal variations in ice surface velocity were limited over the observational period (Figure 5) with ice surface velocity showing no major changes across each of the Adams, Bond West, and Underwood Glaciers. Whilst ice surface velocity was seen to increase by 12% across the FT of Anzac Glacier between 2009 and 2021 (Figure 5c), this velocity increase was not
deemed notable, with the absolute value of acceleration being smaller than the associated error. However, a clear 31% increase was observed at Vanderford Glacier, with ice surface velocity at box IN recorded to accelerate from 201 ± 39 to 264 ± 6 m yr$^{-1}$ between 2000 and 2013 (Figure 5a). Significant variations were also seen at Bond East Glacier, with a period of deceleration measured between 2006 and 2009, followed by a subsequent acceleration between 2009 and 2021 (Figure 5d). Such variation in ice surface velocity was consistent across each of the IN, GL, and FT boxes. At the GL box, ice surface
velocity slowed by 15%, decreasing from 508 ± 42 m yr$^{-1}$ in 2006 to 430 ± 66 m yr$^{-1}$ in 2009 (Figure 5d). Subsequent acceleration was also calculated at 15%, with ice surface velocity observed to increase to 496 m yr$^{-1}$ in 2021 (Figure 5d).



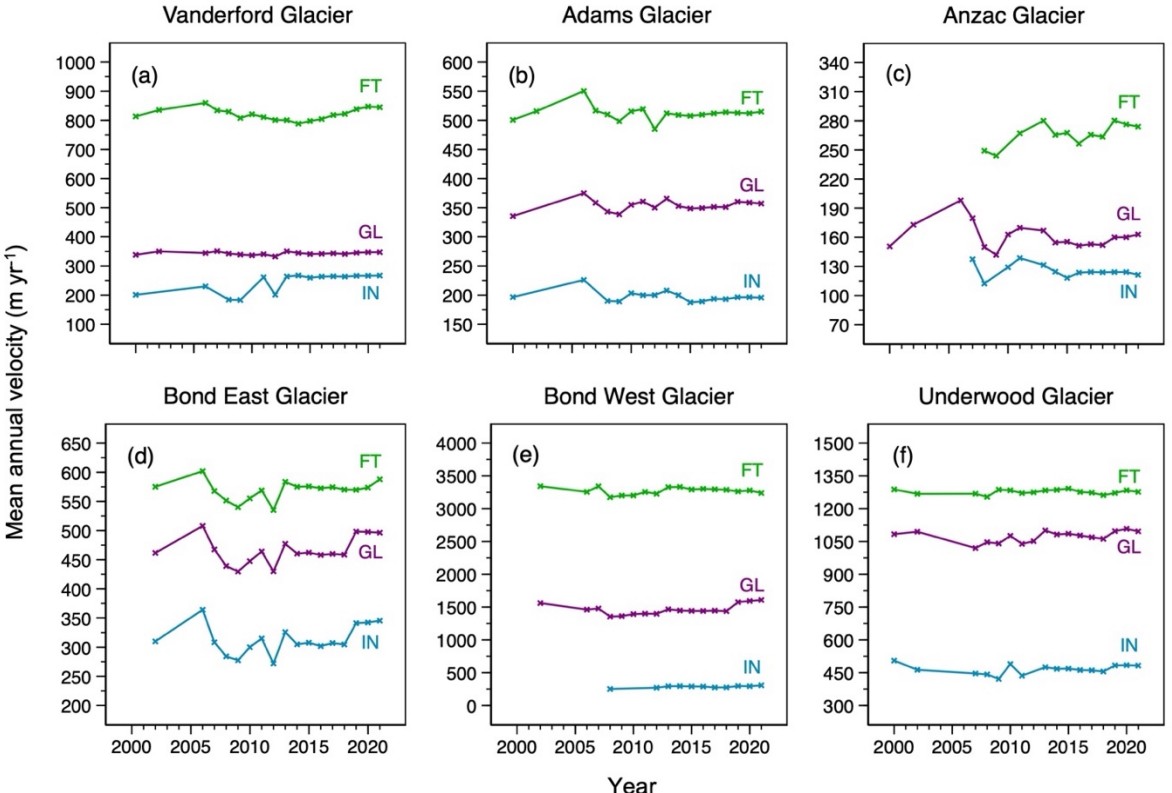

**Figure 5. Mean annual velocity extracted within the inland (IN), grounding line (GL) and floating tongue (FT) boxes across (a) Vanderford Glacier, (b) Adams Glacier, (c) Anzac Glacier, (d) Bond East Glacier, (e) Bond West Glacier, and (f) Underwood Glacier. Velocity data extracted from the ITS_LIVE velocity mosaics between 2000 and 2018 (Gardner et al., 2022) and ENVEO datasets between 2019 and 2021 (ENVEO, 2021). Note the different scales on the y-axes.**

### 3.3. Ice Surface Elevation

The ice surface elevation data provided by both Schröder et al. (2019) and Nilsson et al. (2022) were associated with high levels of uncertainty prior to 2003, particularly during the early 1990s (Figure 6). However, with the exception of Anzac Glacier (Figure 6c), there was a general agreement between each respective dataset regarding the overall pattern of ice surface elevation change measured across each GL box. A clear and consistent thinning trend was measured at Vanderford Glacier throughout the observational period (Figure 6a). Between 2003 and 2017, Schröder et al. (2019) and Nilsson et al. (2022) observed thinning at an average rate of -0.12 and -0.07 m yr[-1], respectively. However, Nilsson et al. (2022) observed an enhanced rate of thinning between 2017 and 2020, measured at -0.22 m yr[-1] . Despite their observed stability between 2003 and 2017, Nilsson et al. (2022) also measured enhanced rates thinning rates across the Adams, Anzac, and Underwood Glaciers between 2017 and 2020 (Figure 6), calculated at an average rate of -0.32, -0.44, and -0.38 m yr[-1], respectively. In





contrast, ice surface elevation was observed to be stable across both Bond East and Bond West between 2003 and 2020, with

Nilsson et al. (2022) measuring minor thickening at an average rate of just +0.01 m yr⁻¹. The patterns of surface elevation

change were observed to be consistent across both the GL (Figure 6) and IN (Figure S2) boxes at each respective glacier.

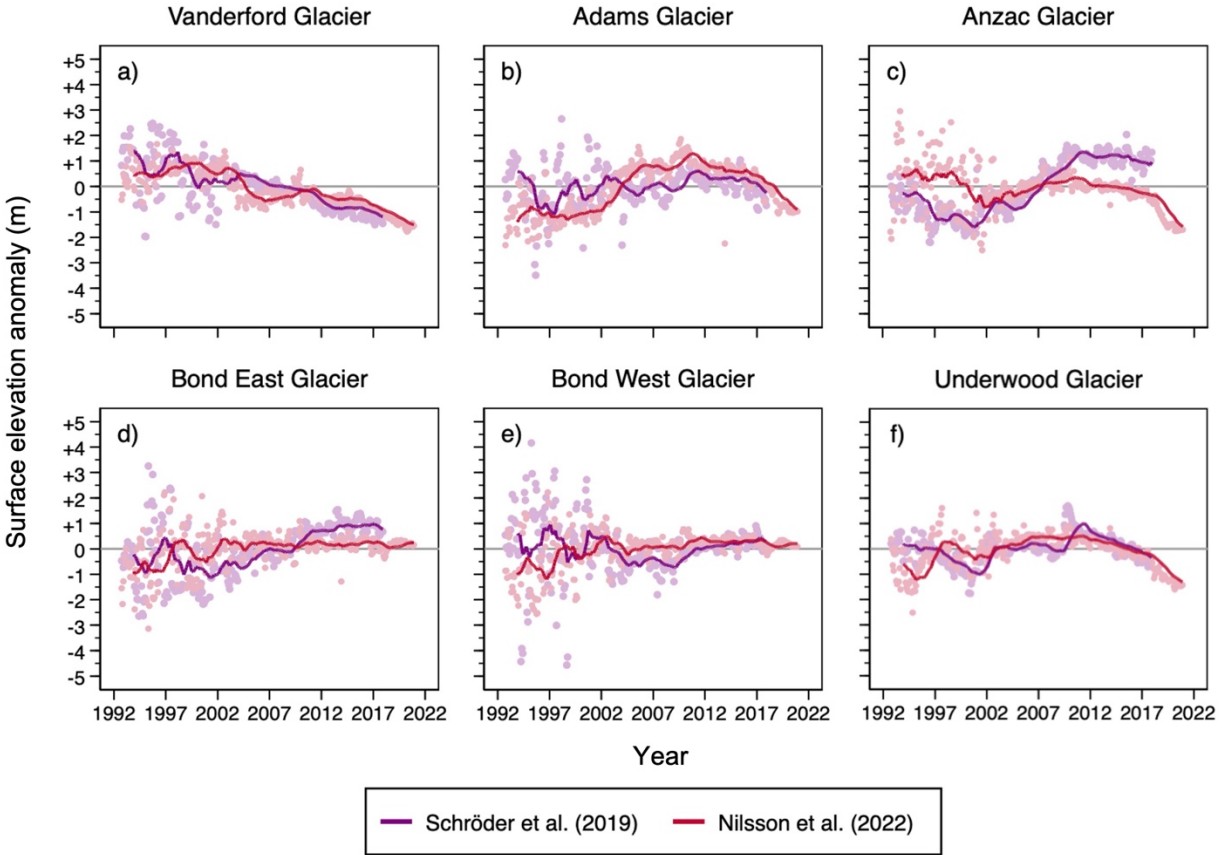

**Figure 6. Monthly surface elevation anomalies observed in each GL box at (a) Vanderford Glacier, (b) Adams Glacier, (c) Anzac**
**Glacier, (d) Bond East Glacier, (e) Bond West Glacier, and (f) Underwood Glacier between 1992 and 2020. Elevation anomalies**
**are calculated relative to the long-term 1992-2017 mean. Bold lines represent 24-month rolling means.**

These trends in ice surface elevation change were also reflected in the dataset produced by Smith et al. (2020) (Figure 7).

Whilst Vanderford, Adams, Anzac and Underwood Glaciers showed an overall decrease in ice surface elevation between

2003 and 2019, both the Bond East and Bond West Glaciers exhibited minor thickening across the same time period. Such

minor thickening was concentrated within ~ 35 km of the coastline, with ice surface elevation comparatively stable inland of

the Bond East and Bond West flowlines (Figure 7).



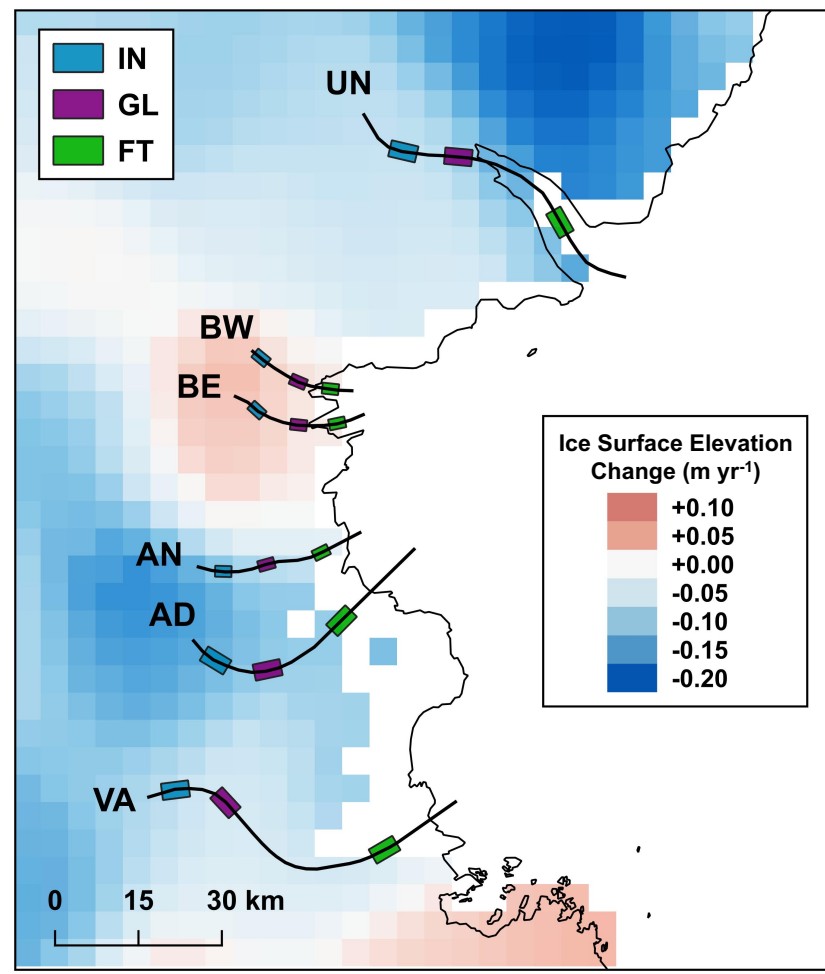

**Figure 7. Rate of ice surface elevation change observed inland of Vincennes Bay between 2003 and 2019, as calculated by Smith et al. (2020). Coastline, shown in black, downloaded from the SCAR Antarctic Digital Database, accessed using Quantarctica 3 (Matsuoka et al., 2021).**

## 3.4. Grounding Line Position

Vanderford Glacier showed the greatest and most consistent grounding line retreat across the observational period, retreating ~ 18.6 km between 1996 and 2020, at an average rate of -0.8 km yr$^{-1}$ (Figure 8a). Such grounding line retreat primarily occurred down a retrograde bedrock slope, but the most recent observed grounding line position appears to be situated on a stabilising ridge (Figure 9a). Between 2016 and 2020, the average rate of grounding line retreat increased to 1.0 km yr$^{-1}$, with ~4.1 km of grounding line retreat measured over this 4-year period (Figure 8a).
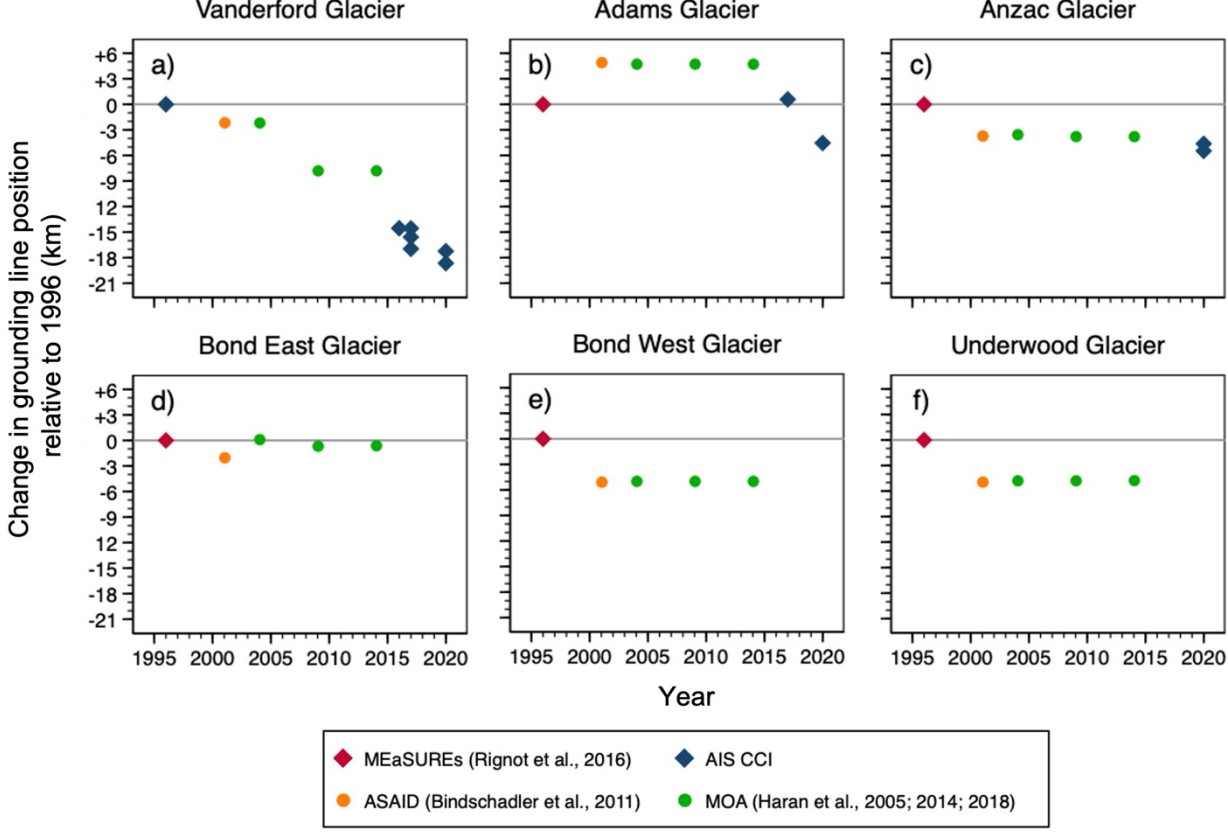

**Figure 8. Change in grounding line position measured relative to the minimum observed 1996 position at (a) Vanderford Glacier, (b) Adams Glacier, (c) Anzac Glacier, (d) Bond East Glacier, (e) Bond West Glacier, and (f) Underwood Glacier. Note that circles represent grounding line positions manually derived from optical imagery, whilst diamonds represent grounding line products derived using DInSAR.**

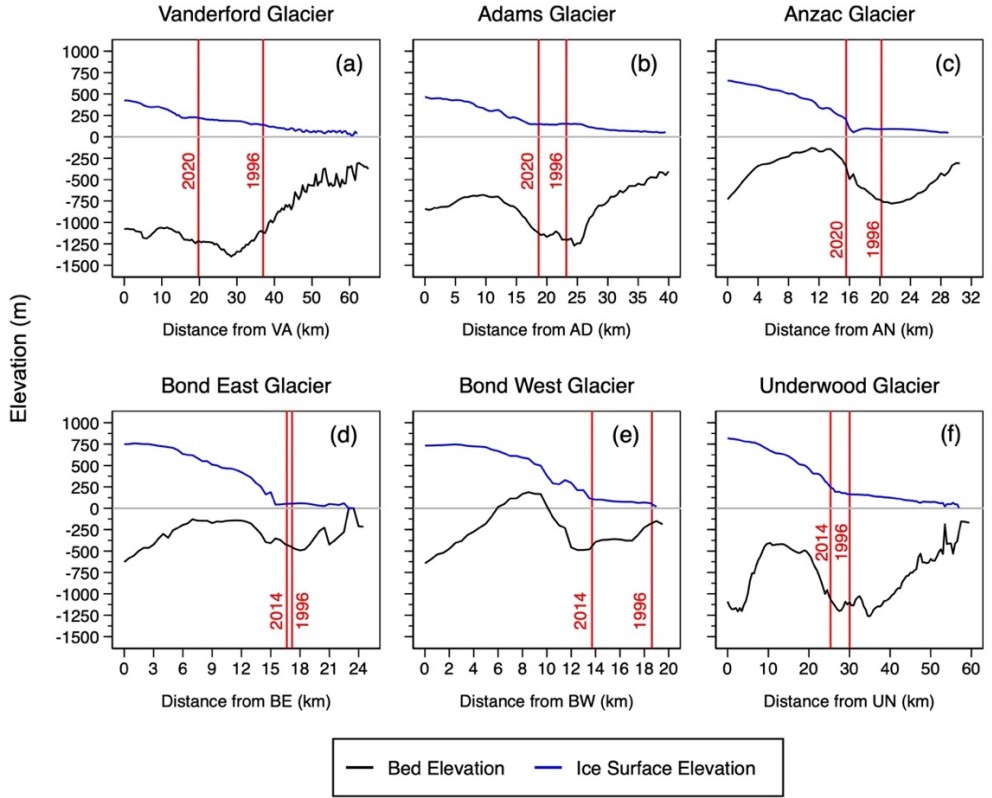

**Figure 9. Ice surface (Howat et al., 2019) and bedrock elevation (Morlighem, 2020) profiles extracted along the central flowlines of (a) Vanderford Glacier, (b) Adams Glacier, (c) Anzac Glacier, (d) Bond East Glacier, (e) Bond West Glacier, and (f) Underwood Glacier. Dated vertical red lines represent the oldest and most recent grounding line positions observed respectively.**


In contrast, the grounding line positions of the Adams, Anzac, Bond East, Bond West, and Underwood Glaciers were observed to be stable between 2001 and 2014, with the ASAID and MOA grounding line products digitised in nearly identical locations (Figure 8). Despite an initial grounding line advance of ~ 5 km between 1996 and 2001, an overall retreat

of ~ 4.5 km was observed at Adams Glacier between 1996 and 2020 (Figure 8b). A similar overall pattern of grounding line retreat was observed at Anzac Glacier, measured at ~ 5.4 km over the same time period (Figure 8c).

Between 1996 and 2014, Bond West and Underwood Glacier exhibited very similar patterns of grounding line retreat, retreating ~4.9 and ~4.7 km, respectively (Figure 8). The majority of this retreat was observed between 1996 and 2001.

Unlike the other outlet glaciers within Vincennes Bay, Bond East's grounding line was seen to be stable across the observational period (Figure 8d). Between 1996 and 2001, the grounding line was estimated to have retreated ~2 km. However, between 2004 and 2014, Bond East's grounding line remained within ~600 m of the observed 1996 position. It should be noted that more recent grounding line positions were not available at the Bond East, Bond West, and Underwood Glaciers.




## 4. Discussion

### 4.1. Recent dynamic change and future evolution of Vanderford Glacier

Our results show extensive grounding line retreat at Vanderford Glacier, measured at 18.6 km between 1996 and 2020 (Figure 8a). This corresponds to an average rate of retreat of 0.78 km yr$^{-1}$, representing the fastest decadal-scale grounding

line retreat reported in East Antarctica (Stokes et al., 2022). Such retreat is consistent with that seen at Thwaites Glacier, measured at 0.8 km yr$^{-1}$ between 1992 and 2017 (Milillo et al., 2019) and widely considered to result from the enhanced intrusion of warm mCDW across the continental shelf towards sub ice-shelf cavities (Thoma et al., 2008; Steig et al., 2012; Paolo et al., 2015; Turner et al., 2017a; Scambos et al., 2017; Rignot et al., 2019). The marked grounding line retreat observed at Vanderford Glacier is therefore consistent with the notion that warm mCDW is able to access local ice-shelf

cavities below the Vanderford Ice Shelf, driving high rates of basal melting (Depoorter et al., 2013; Ribeiro et al., 2021).

The rapid rate of grounding line retreat measured at Vanderford Glacier is anomalous within Vincennes Bay, calculated to be nearly four times greater than the average rate of retreat (0.20 km yr$^{-1}$) recorded across the Adams, Anzac, Bond East, Bond West, and Underwood Glaciers. With mCDW typically observed to access sub ice-shelf cavities via deep subglacial troughs

(Jenkins et al., 2010; Scambos et al., 2017; Rignot et al., 2019), this significant difference may be indicative of a bathymetric pathway that favours preferential intrusion of mCDW towards the Vanderford Ice Shelf, rather than the other studied glaciers. Bathymetric data remains limited across Vincennes Bay, but recent echosounding carried out onboard RSV *Nuyina* revealed an undiscovered canyon at the front of Vanderford Glacier, estimated to be more than 55 kilometres in length, reaching a maximum depth of 2200 m (Figure 10) (Australian Antarctic Division, 2022). This canyon might provide a

potential pathway for the incursion of mCDW at depth towards the Vanderford Ice Shelf, thereby facilitating enhanced rates of grounding line retreat relative to the other Vincennes Bay outlet glaciers. In addition, the anomalously high rate of grounding line retreat observed at Vanderford Glacier may be attributed to the underlying bedrock geometry. Figure 9a indicates that between 1996 and 2020, the majority of Vanderford's grounding line retreat occurred down a retrograde slope. In contrast, with the exception of Bond West Glacier (Figure 9e), the grounding line retreat observed at other glaciers within

Vincennes Bay generally occurred along prograde slopes. As retrograde bedrock slopes favour more extensive grounding line retreat for a given basal melt rate (Milillo et al., 2019; Millan et al., 2022), the high rate of grounding line retreat seen at Vanderford Glacier may hence also be the product of the underlying bedrock geometry.





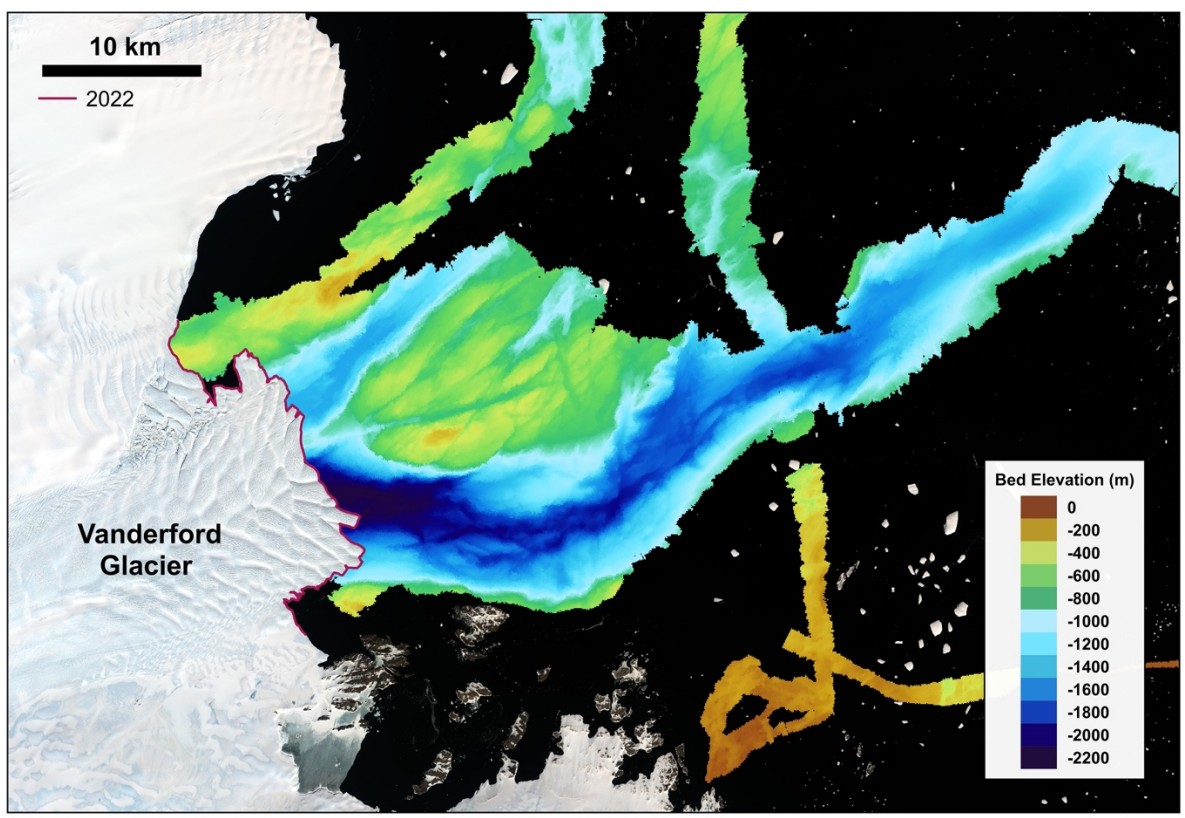

**Figure 10. Bathymetry mapped at the front of Vanderford Glacier, collected during RSV *Nuyina's* maidan voyage to Antarctica**
**(Commonwealth of Australia, 2022). Background satellite imagery was collected in February 2022, with red line showing the**
**associated digitised terminus position.**

The intrusion of warm mCDW to sub ice-shelf cavities facilitates the enhanced oceanic thinning of floating ice shelves
(Paolo et al., 2015). Such ice shelves transmit a critical buttressing force upstream, restraining the outflow of grounded ice
(Fürst et al., 2016). The thinning of ice shelves reduces this buttressing potential, often triggering dynamic thinning and
acceleration across upstream glaciers (Reese et al., 2018). For example, Thwaites Glacier accelerated by 33% between 2006
and 2013 (Mouginot et al., 2014), with dynamic thinning of at least 1.5 m yr⁻¹ measured between 2012 and 2020 (Bevan et
al., 2021). Whilst our results reveal a 31% increase in inland velocity at Vanderford Glacier between 2000 and 2013,
velocity was seen to be stable across the grounding line throughout the observational period (2000 – 2021), with no
significant acceleration recorded (Figure 5a). In addition, although a consistent decrease in surface elevation was measured
across Vanderford Glacier (Figure 6a) the average rate of ice thinning, calculated to range between -0.07 and -0.12 m yr⁻¹
(2003 – 2017), was an order of magnitude lower than recorded at other glaciers undergoing similar rapid grounding line
retreat within the Amundsen Sea Embayment, such as Thwaites Glacier (Pritchard et al., 2009; Flament & Rémy, 2012;
Konrad et al., 2017). This could be attributed to variations in surface mass balance, with a comparatively higher rate of



snowfall within the Aurora subglacial basin potentially obscuring a dynamic thinning signal. However, the rate of surface lowering measured across the neighbouring Totten Glacier (Figure 1b) was also significantly higher than at Vanderford Glacier, estimated up to -1.7 m yr$^{-1}$ between 2003 and 2008 (Khazendar et al., 2013). This therefore suggests that the comparatively moderate rate of thinning seen across Vanderford Glacier is not the product of dynamic thinning being mitigated by high regional snowfall. Rather, the rate of dynamic thinning observed across Vanderford Glacier appears to

have been significantly lower. This indicates that the ocean-driven reduction in buttressing force exerted by the Vanderford Ice Shelf has presently been limited. Nonetheless, Fürst et al. (2016) calculated that just 20.2% of the Vanderford Ice Shelf is categorised as passive, i.e. can be removed without initiating a dynamic glaciological response. It may thus be predicted that continued intrusion of warm mCDW will likely reduce the buttressing force exerted by the Vanderford Ice Shelf, potentially initiating acceleration and enhanced dynamic thinning across Vanderford Glacier in the coming decades.


Whilst Vanderford Glacier is currently grounded on a stabilising bedrock ridge (Figure 9a), extension of the central flowline shows that retrograde slopes are observed inland along the Vanderford Trench, with a significant decrease in bedrock elevation seen approximately 70 km inland of the present grounding line position (Figure 11c). If the current rate of grounding line retreat (0.78 km yr$^{-1}$) was to continue, this steep bedrock slope would be reached within 100 years, potentially

triggering MISI. Indeed, using the BISICLES adaptive mesh ice-sheet model, Sun et al. (2016) simulated Vanderford's grounding line to retreat rapidly along the Vanderford Trench, separating Law Dome (Figure 1b) from the continental ice sheet within 1000 years. However, Sun et al. (2016) also emphasised that such extensive grounding line retreat is dependent on the future oceanic forcing, requiring basal melt rates elevated above those observed at present.







**Figure 11. Extended flowline digitised along the main flow of Vanderford Glacier overlain on maps of a) ice velocity extracted from the 2018 ITS_LIVE ice velocity mosaic (Gardner et al., 2022), and b) bedrock topography extracted from BedMachine (Morlighem, 2020). c) Bed elevation sampled from BedMachine (Morlighem, 2020) along the extended flowline. 1996 MEaSUREs (Rignot et al., 2016) and 2020 AIS CCI grounding line positions are displayed in red and purple respectively.**



Recent oceanographic observations suggest that the oceanic heat supply to East Antarctica is increasing (Herraiz-Borreguero & Garabato, 2022), potentially facilitating such increased basal melt rates. The mid-depth CDW found along the continental slope off East Antarctica has warmed significantly since the 1990s, with an increase of 0.29°C per decade estimated near Vincennes Bay (105 °E – 111°E). This warming has been attributed to the southward shift of the Antarctic Circumpolar Current (ACC) (Yamazaki et al., 2021), understood to have been driven by a poleward shift of the westerlies over the

Southern Ocean associated with summertime positive trends in the southern annular mode (Herraiz-Borreguero & Garabato, 2022). With climatic models predicting that the southern annular mode will continue trending towards its positive phase under high-emission scenarios (Zheng et al., 2013; Lim et al., 2016; Lee et al., 2021), further CDW warming may be expected off East Antarctica, enabling enhanced basal melt rates (Herraiz-Borreguero & Garabato, 2022). Subsequent increased outflow of glacial meltwater will likely further hinder the formation of DSW in the Vanderford polynya, thereby

facilitating the enhanced incursion of warm mCDW at depth (Ribeiro et al., 2021). This positive feedback could drive rapid grounding line retreat at Vanderford Glacier, potentially providing the oceanic forcing required to initiate MISI (Sun et al., 2016).

### 4.2. Decadal patterns of terminus position change observed across the Vincennes Bay outlet glaciers potentially
**correlated to sea ice production**

The decadal variations in terminus position observed across Vincennes Bay (Table 3) are seen to correspond closely with the wider decadal-scale patterns reported along the Wilkes Land coast (Miles et al., 2016) (Figure 12). The majority of Wilkes Land outlet glaciers were seen to retreat between 1974-1990, switch to a period of sustained advance between 1990-2000, before retreating again between 2000-2012 (Miles et al., 2016). Miles et al. (2016) suggested that such decadal fluctuations

between terminus retreat and advance are strongly correlated to rates of sea-ice production around East Antarctica. They noted that enhanced brine rejection from increased sea-ice production can result in destratification of the water column (Petty et al., 2014), facilitating the formation of DSW and hence production of Antarctic Bottom Water (AABW) (Kusahara et al., 2011). Representing the densest water mass in the ocean (Ribeiro et al., 2021), AABW is understood to inhibit intrusion of mCDW, thereby suppressing basal melting and facilitating terminus advance (Miles et al., 2016). In contrast, decreased sea-

ice production leads to a reduction in DSW formation and hence decreased production of AABW. This allows enhanced intrusion of mCDW, driving basal melting and thus terminus retreat (Miles et al., 2016). Whilst detailed analysis of decadal trends in sea ice production is yet to be carried out within Vincennes Bay, Miles et al. (2016) noted a negative correlation between sea-ice production and average air temperatures over the sea-ice production season (April to October) along the Wilkes Land coast.




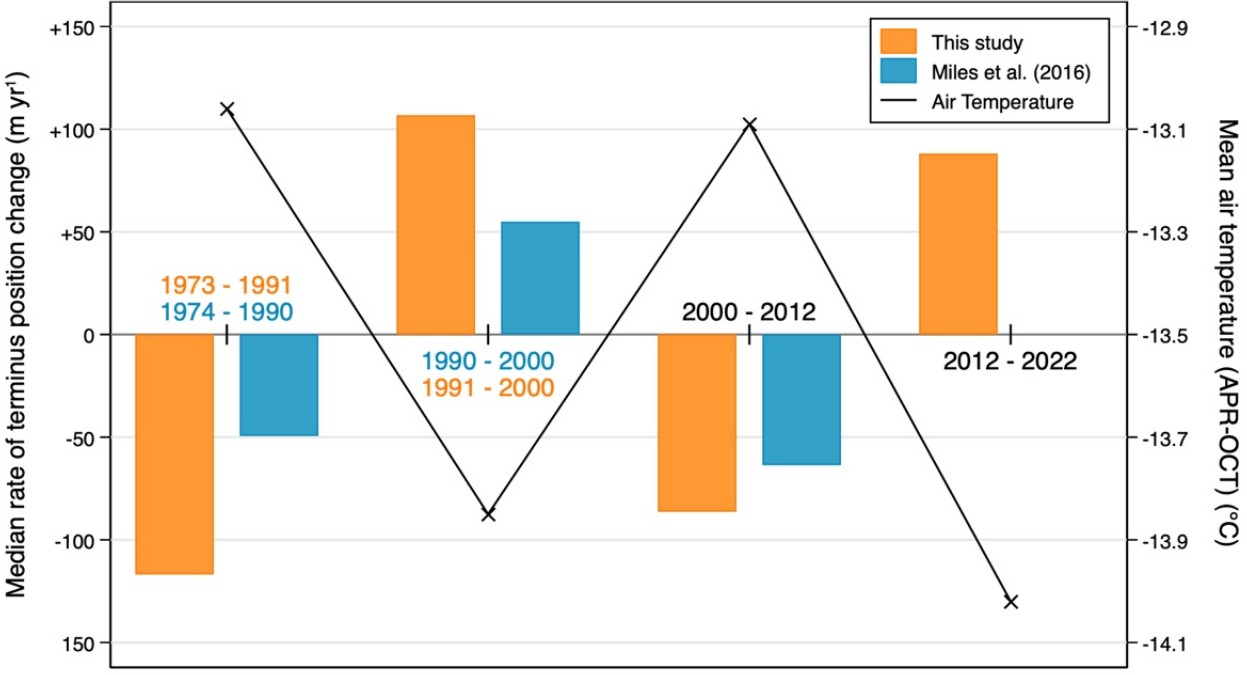

**Figure 12. Median decadal terminus position change reported in this study within Vincennes Bay, shown in orange, in comparison to that observed along the Wilkes Land coastline by Miles et al. (2016), shown in blue. This study (n = 6), Miles et al. (2016) (n= 15 (1974-1990), 37 (1990-2000), 39 (2000-2012)). Mean surface air temperatures measured at Casey Station (66.3°S 110.5°E) over the sea ice production season (April to October), extracted from Met-READER (2022), displayed in black.**

To consider this potential correlation within Vincennes Bay, monthly surface air temperature data collected at Casey Station (Figure 1b) were downloaded and analysed (Met-READER, 2022). The results indicated such a negative correlation was also observed within Vincennes Bay, with higher temperatures observed to be coincident with periods of terminus retreat (Figure 12). The decrease in mean surface air temperature and overall pattern of terminus advance observed between 2012-2022 (Figure 12) may therefore be indicative of increased sea ice production within Vincennes Bay over this most recent decade.

### 4.3. Enhanced thinning of the Vanderford, Adams, Anzac, and Underwood Glaciers observed between 2017 and 2020

Analysis of ice surface elevation shows a pattern of accelerated thinning across each of the Vanderford, Adams, Anzac, and Underwood Glaciers between 2017 and 2020 (Figure 6). The most significant acceleration was measured at Anzac Glacier, with the average rate of thinning increasing from -0.01 m yr$^{-1}$ between 2003 and 2017 to -0.44 m yr$^{-1}$ between 2017 and 2020 (Table 4) (Nilsson et al., 2022).





| | Average rate of thinning (m yr$^{-1}$) | | | |
| --- | --- | --- | --- | --- |
| | **Vanderford** | **Adams** | **Anzac** | **Underwood** |
| **2003 - 2017** | -0.07 | -0.02 | -0.01 | -0.03 |
| **2017 – 2020** | -0.22 | -0.32 | -0.44 | -0.38 |

**Table 4. The average rate of thinning observed within the GL box of the Vanderford, Adams, Anzac, and Underwood Glaciers between 2003-2017 and 2017-2020, respectively. Note that these rates of thinning were extracted from the ice surface elevation dataset provided by Nilsson et al. (2022).**

Whilst the observational period is relatively short, this synchronous dynamic response may potentially be indicative of a
common external forcing. With the negative correlation between sea ice production and the intrusion of mCDW previously outlined (Miles et al., 2016), such enhanced thinning could be related to the significant decline in Antarctic sea ice extent seen during the austral spring of 2016 (Turner et al., 2017b). Whilst negative sea ice anomalies were observed to be transient across the Western Pacific Ocean (90°E – 160°E) (Turner et al., 2017b), decreased production of sea ice may have facilitated the increased incursion of warm mCDW within Vincennes Bay, thereby driving enhanced basal melting and dynamic
thinning across the studied outlet glaciers. With a satellite-era record minimum Antarctic sea ice extent observed in February 2022 (Turner et al., 2022), such accelerated thinning may be expected to continue in the immediate future.

In addition, the significant decline in Antarctic sea ice extent observed in 2016 (Turner et al., 2017b) may have had potentially important implications for the stability of the glacier termini. Previous analysis of outlet glacier dynamics within
Porpoise Bay revealed that large calving events observed in January 2007 and March 2016 were linked to the break-up of multi-year landfast sea ice (Miles et al., 2017). Miles et al. (2017) therefore emphasised that sea ice concentrations can exert a significant control on terminus stability, supporting correlations between terminus change and sea ice mélange conditions previously observed in Greenland (Reeh et al., 2001; Moon et al., 2015). The negative sea ice anomalies seen in 2016 (Turner et al., 2017b) may thus have reduced the stability of the outlet glacier termini, facilitating terminus advance and
associated longitudinal thinning across the grounding line. Indeed, analysis of terminus position change indicates that each of the Vanderford, Adams, and Underwood glaciers underwent a period of sustained advance between 2017 and 2020 (Figure 2). However, Anzac Glacier was recorded to be comparatively stable between 2017 and 2020, retreating by just -86 m (Figure 2). This suggests the enhanced rate of thinning measured across Anzac Glacier was not the product of longitudinal thinning. It may thus be inferred that the thinning instead occurred in situ, perhaps further supporting the notion of increased
basal melt rates driven by enhanced mCDW intrusion.



### 4.4. High velocity at Bond West Glacier

Ice discharge across the grounding line is dependent on ice velocity (Moon et al., 2015); the accurate assessment of ice velocity is hence crucial for determining sea level contributions. In comparison to the other outlet glaciers within Vincennes
Bay, Bond West Glacier flows at a significantly higher velocity (Figure 4). The fastest flowing outlet glacier in East Antarctica, Shirase Glacier, is measured to reach speeds in excess of 2,200 m yr$^{-1}$ across the grounding line (Miles et al. 2022), with a maximum speed of 2,700 m yr$^{-1}$ measured at the calving front (Pattyn & Derauw, 2002). Whilst Bond West flows more slowly across its grounding line, at an average velocity of 1,463 m yr$^{-1}$ (2002 – 2021), a more extreme acceleration is observed  across Bond West's floating tongue, with a maximum flow speed of 3,344 m yr$^{-1}$ measured in 2002
(Figure 5e). The floating tongue is thus heavily fractured (Figure 3e), showing a crevasse pattern more typically observed across fast-moving Greenlandic glaciers, such as Jakobshavn Isbræ (Mayer & Herzfeld, 2000). The high ice surface velocity measured across Bond West's floating tongue may be a product of the underlying bedrock topography, with a steep slope of 9.6° measured immediately upstream of the grounding line (Figure 9e). In addition, fast-flowing upstream ice is observed to converge through a comparatively narrow ice shelf, constrained to the east by a stable ice rise and to the west by elevated
bedrock topography (Figure 1b). Enhanced ice velocity must thus occur in order to maintain constant ice discharge through a smaller cross-sectional area (Winsborrow et al., 2010).

### 4.5. Importance of accurate DInSAR grounding line mapping

The accurate mapping of grounding line positions is fundamental for understanding ice sheet mass balance and assessing
potential future contributions to sea level rise (Rignot et al., 2014; Mohajerani et al., 2021). However, Antarctic grounding line positions have been mapped using a variety of different methods, with the term 'grounding line' often being used to refer to different distinct features across the grounding zone (Bindschadler et al., 2011). The most accurate mapping technique is typically considered to be DInSAR, with grounding line positions derived precisely through the analysis of tidally induced vertical motion (Rignot et al., 2014; Li et al., 2021; Mohajerani et al., 2021). In contrast, the manual
digitisation of grounding line positions from optical imagery, delineated using the most seaward observed break-in-slope, can be associated with high levels of uncertainty (Fricker et al., 2009; Rignot et al., 2011), particularly at fast-flowing outlet glaciers such as those found within Vincennes Bay (Bindschadler et al., 2011; Christie et al., 2016). This has been attributed to the notion that fast-flowing ice streams are typically moving via basal sliding, meaning the transition to zero basal resistance across the ice shelf therefore produces a less marked break-in-slope (Rignot et al., 2011). With the exception of
Vanderford Glacier, the results presented in this study show that the grounding line positions mapped from optical imagery (ASAID and MOA datasets) are located in near-identical locations at each of the Vincennes Bay outlet glaciers between 2001 and 2014 (Figure 8). Whilst this may be indicative of grounding line stability, the lack of precise DInSAR grounding line mapping over this time period precludes such inferences being made with certainty. We therefore propose that such



DInSAR techniques should be prioritised in order to facilitate accurate grounding line mapping across such potentially dynamic regions of East Antarctica.

## 5. Conclusions

This study provides the first detailed investigation of ice dynamics observed across the outlet glaciers of Vincennes Bay, a region recently recorded to have the warmest intrusions of mCDW within East Antarctica (Ribeiro et al., 2021). Our results confirm extensive grounding line retreat at Vanderford Glacier, measured at 18.6 km between 1996 and 2020, representing

an average retreat rate of 0.8 km yr$^{-1}$. This reflects the highest rate of grounding line retreat reported for any glacier within East Antarctica and is consistent with the notion that warm mCDW is able to intrude at depth, accessing cavities formed below the Vanderford Ice Shelf and driving high rates of basal melt (Depoorter et al., 2013; Ribeiro et al., 2021). Whilst currently grounded on a stabilising bedrock ridge, retrograde slopes are observed inland along the Vanderford Trench. If grounding line retreat continues at the present rate, this retrograde slope will be reached within 100 years, raising the

potential for MISI. With basal melt predicted to further inhibit the formation of DSW within the Vanderford polynya (Ribeiro et al., 2021; Herraiz-Borreguero & Garabato, 2022), mCDW intrusion may be enhanced, thereby providing the oceanic forcing required to drive further grounding line retreat. Although the dynamic response of Vanderford Glacier has been limited thus far, a consistent thinning trend was seen over the observational period, measured between -0.07 and -0.12 m yr$^{-1}$ (2003-2017). Ocean forcing may be expected to enhance this dynamic response, with both accelerated thinning and

increased ice surface velocities predicted across Vanderford Glacier over the coming decades.

The study also reveals that decadal changes in frontal position measured across the Vincennes Bay outlet glaciers correspond closely with wider patterns reported along the Wilkes Land coastline (Miles et al., 2016). Analysis of air surface temperature data collected at Casey Station supports the notion that such trends may be correlated to variable sea ice production. It is

suggested that decreased sea-ice production limits the formation of DSW, thereby facilitating the enhanced intrusion of mCDW at depth (Miles et al., 2016). An accelerated thinning signal measured across each of the Vanderford, Adams, Anzac, and Underwood Glaciers between 2017 and 2020 could be attributed to this feedback, with the preceding widespread decline in sea ice extent seen across Antarctica (Turner et al., 2017b) potentially facilitating enhanced mCDW intrusion and increased basal melting. This decline in sea ice extent may have also decreased the stability of the outlet glacier termini, with

the Vanderford, Adams, and Underwood Glaciers each observed to advance between 2017 and 2020. Such terminus advance is suggested to have driven longitudinal thinning, further contributing to the enhanced thinning signal observed.

In comparison to Vanderford Glacier, the extent of grounding line retreat observed across the Adams, Anzac, Bond East, Bond West, and Underwood Glaciers was significantly lower, averaged at a rate of 0.2 km yr$^{-1}$. With a recently discovered

canyon providing a potential bathymetric pathway for mCDW intrusion towards Vanderford Glacier, the comparatively





lower rates of grounding line retreat observed across the remaining Vincennes Bay outlet glaciers may be indicative of a relative lack of bathymetric pathways towards these termini. Indeed, the grounding line positions of the Adams, Anzac, Bond East, Bond West, and Underwood Glaciers were seen to be stable between 2001 and 2014. However, such stability was inferred from datasets derived through manual digitisation of the most seaward observed break-in-slope and are hence

associated with higher levels of uncertainty. The results of this study thus emphasise the need to prioritise precise mapping of grounding line positions using DInSAR techniques, particularly across dynamic regions, such as Wilkes Land.

## 6. Data Availability

The Landsat and Sentinel imagery used in this study are available from the United States Geological Survey EarthExplorer

(https://earthexplorer.usgs.gov/). The manually digitised terminus positions are available upon request. The ITS_LIVE annual velocity mosaics (2000-2018) are available from NASA's National Snow and Ice Data Center (NSIDC) (https://doi.org/10.5067/6II6VW8LLWJ7), whilst the ENVEO monthly velocity mosaics (2018-2020) are available from the NERC EDS Centre for Environmental Data Analysis (http://dx.doi.org/10.5285/00fe090efc58446e8980992a617f632f). The monthly surface elevation change datasets produced by Schröder et al. (2019) (1978-2017) and Nilsson et al. (2022) (1985-

2020) are available from (https://doi.org/10.1594/PANGAEA.897390) and (https://doi.org/10.5067/L3LSVDZS15ZV). The ice-column thickness-change-rate estimates (2003-2019) made by Smith et al. (2020) are available from the ResearchWorks Archive (http://hdl.handle.net/1773/45388). The AIS CCI grounding line positions derived in this study are available from the ENVEO cryoportal (cryoportal.enveo.at). The MEaSUREs (Rignot et al., 2016) 1996 grounding line positions are available from NASA's NSIDC at (https://doi.org/10.5067/IKBWW4RYHF1Q). The MOA grounding line positions from

2004 (Haran et al., 2005), 2009 (Haran et al., 2014), and 2014 (Haran et al., 2018) are also available from NASA's NSIDC at (https://doi.org/10.5067/68TBT0CGJSOJ), (https://doi.org/10.5067/4ZL43A4619AF), and (https://doi.org/10.5067/RNF17BP824UM), respectively. The ASAID grounding line dataset (1999-2003) is available from the U.S. Antarctic Program Data Center (http://dx.doi.org/10.7265/N56T0JK2). The monthly surface air temperature collected from Casey Station is available from (https://legacy.bas.ac.uk/met/READER/surface/Casey.All.temperature.html).





## 7. Supplement

## 8. Author Contributions

HP, CS, and SJ designed the initial study. HP undertook the data collection and conducted the analysis, with guidance from CS and SJ. DF and LK completed the DInSAR grounding line mapping within Vincennes Bay, with LK writing the associated methods section. HP led the remaining manuscript writing, with input from all authors.

## 9. Competing Interests

CRS is a member of the editorial board of *The Cryosphere*.

## 10. Acknowledgements

CRS and SSRJ acknowledge funding from the Natural Environment Research Council (NE/R000824/1).

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
