# Peer review of "Extensive and anomalous grounding-line retreat at Vanderford Glacier, Vincennes Bay, Wilkes Land, East Antarctica."

_The Cryosphere, 2022_

## Referee Comment (RC2)

**Review of tc-2022-217 "Extensive and anomalous grounding line retreat at Vanderford Glacier, Vincennes Bay, Wilkes Land, East Antarctica" by Hannah Picton et al., 2023**

In this manuscript, Picton et al. use a range of remotely sensed datasets to examine recent glaciological changes within Vincennes Bay, East Antarctica. Amongst other interesting indicators of glacier change, they report, most notably, upon the rapid retreat of Vanderford Glacier's grounding line at a rate of 0.8 km/yr between 1996 and 2020.

Overall, the paper is generally well-presented, well-structured and is scientifically robust, and even considering the observations of Vanderford Glacier's behavior alone – which places it as the 4[th] fastest retreating glacier in Antarctica over the satellite era – I believe this manuscript will be of broad interest to the readership of *The Cryosphere*. For this reason, I recommend publication. Prior to publication, however, I believe the manuscript has several limitations in its current form which should be addressed. These limitations are detailed in my comments below.

**General Comments**

*Vanderford grounding-line retreat.* While the retreat rate reported here is undoubtedly significant, the problem is that it is for the most part not a new finding. This is because, as the authors themselves allude to on Line 84, Rignot et al. (2019) have previously reported upon this behavior as observed between 1996 and 2017 (over which time they also find a retreat rate of 0.8 km/yr). In this regard there are two key issues with the manuscript in its current form:

1) The grounding-line retreat-related findings – as presently reported at least – are perhaps more incremental than the narrative of the manuscript would suggest.

2) Apart from the introduction (Line 84), no further acknowledgement of Rignot et al.'s earlier observations is included, which could be misconstrued by some as slightly disingenuous.

To remedy these issues, I would suggest the authors:
1) Rework the text to contextualize their findings more clearly alongside this earlier research. (In e.g. the abstract and conclusion, phrasing like "Our results confirm extensive grounding-line retreat…" is used, but this doesn't make it explicit that this is a confirmation of a previously documented observation). More explicit follow-up discussion of the fact that the author's observations show *continued* retreat since 2017 would also be beneficial, and serve to demonstrate that their research goes beyond that discussed by Rignot et al.

2) What's also new (and arguably much more interesting) relative to the simple trend of 0.8 km/yr reported here and in Rignot et al. (2019) is the seemingly step-wise, temporally variable patterns of retreat observed between 1996 and 2021 (Fig. 8). I think a more explicit/nuanced discussion of this phenomenon and its links with e.g. changes in bed topography/MISI (or otherwise) as seen in Fig. 9 would make for a much more interesting read, while again going beyond that described in Rignot et al. (2019). (See also my *Minor Comment* on Fig.9 below).

*Structure/presentation of manuscript.* While generally well-presented/written overall, I believe the manuscript could also be overhauled in places (abstract, discussion and conclusion especially) to offer a more succinct / 'to the point' discussion of the key points and novel findings only. Perhaps most importantly, I think the structure of the discussion requires some careful refocusing, as at present it contains a lot of unnecessary details which should either be moved to the Methods, Results or Supplementary Information. Elements of the Discussion and Conclusion also have the tendency to jump back and forth between ideas and/or from one sub-section to the next, which I think should be corrected for improved readability. (See my *Minor Comments* below for some examples of these sorts of issues).

***Minor comments***

Line 13 – The ice surface velocity, thinning and GL position datasets are also derived from remotely sensed techniques, so I suggest rephrasing the sentence to better convey this point.

L13 – synthetic aperture radar

L34 – driven > dominated (I agree that these are the two main glaciers, but other parts of the coast are just as sensitive to oceanic influence and are now contributing to these trends too)

L57- 'measured to' > 'having undergone'?

L60 – Weaker easterly winds relative to what and where? This was unclear to me. This line should be revised to clarify this.

Fig.1 – Nice figure. Could 1b be underlain by hillshade and/or contours to make the topography 'pop out' more? In 1a, I also strongly recommend displaying the most recent GL (or ideally all of those included in your timeseries if the figure doesn't look too cluttered) to give the reader an instant sense of how each glacier has retreated through time. To help the reader find the locations referred to in the text (e.g., L81), please also add lat/lon graticules to both panels (and all other maps for that matter).

L101+ – Here, I suggest removing the methodological detail behind the analyses performed (terminus position, velocity, elevation, GL change) as this information is better placed in the following section.

L109 – 'USGS Earth Explorer data repository, with …'

L112 – Why was co-registration of the earlier scenes performed? I presume this pertains to the old ephemerides and DEMs used for geocoding Landsat 1-4 images, and thus the need to ensure spatially consistent imaging through time? Worth stating that here if so. Also, why was co-registration not performed for the ARGON mosaic? Considerable positional errors can exist in those images, and likely exist in the Kim et al. mosaic too.

L119 – … are hereafter referred to as …

L132 – … the AutoRIFT feature tracking algorithm.

L135 – Typo? I believe ENVEO records date back to 2014?

L136 – Suggest rephrasing to say "… using a combination of coherent and incoherent feature tracking techniques". (following Nagler et al. (2021; doi:10.1109/igarss47720.2021.9553514) which should also be cited here).

L184-197 – As the authors are aware, the grounding line and hinge line are two different components of the grounding zone, the latter being a proxy for the former which cannot be detected from satellite imaging. This fact should be stated somewhere in this paragraph, if anything to make it clear that the two components haven't been conflated.

L206/7 – '… was generated using … as applied to ERS-1 and ERS-2 imagery…'.

L216 – Should read '… most seaward, spatially continuous break-in-slope' (since multiple discontinuous breaks-in-slope can exist downstream over, for example, pinning points or ice rumples).

L227 – Following my comments on L184-197, I think it's important to state here that it's difficult to compare break-in-slope and hinge line positions in a direct/like-for-like manner, because they're ultimately measuring two very different components of the grounding zone. Therefore, even given the standalone instrument/technique errors shown in Table 2, any changes in GL position identified from the two techniques should be interpreted with caution, with any further discussion restricted to those exhibiting pronounced retreat where we can be confident change represents a true signal.

L230-234 – I'm not sure I follow this, and specifically why the two GLs would be so far apart. Geocoding issues? Tidal effects (if different imaging dates)? Both? Additional information explaining this would be good here.

Fig.2 – Suggest noting the non-linear x-axis scale in the caption or, even better, editing the figures to show a linear scale.

Fig.3 – Nice figure. Caption should read '…from the USGS Earth Explorer data repository' or similar.

Fig.4 – Nice figure, but what does, for example, 'Distance from VA' mean on the x-axis? Does this literally mean from the 'VA' label on Fig 1? If so, suggest changing the notation to read 'Distance downstream' (or similar), and annotating VA to VA' on both Fig.1 and the cross section profiles of Fig.4.

L290 – Associated errors. This is great, but please also show these errors on Fig 5 too.

Figs. 6 and 7 – These are clear, detail-rich figures. My one thought, however, is whether they could be merged somehow to save the reader flicking back and forth between figs?

L340 – Following my comment on L227, I think it's important to mention here that the retreat observed falls greatly outside sensor error limits, and likely also any between-sensor uncertainties associated with combined DInSAR/break-in-slope comparisons.

Fig 8 – This is a nice figure, although I question whether it's integral to the interpretation presented. I think a revised version of Fig. 9 with each GL position shown would be far more impactful, with Fig. 8 moved to the supplementary information instead. Either way, remove the word 'manually' from the caption since the methods states that both break-in-slope and DInSAR mapping was carried out in this way.

Fig.9 – Please add each GL location to portray the temporal evolution of retreat more clearly, and if needed revise the results/discussion to reflect any temporal patterns of retreat this reveals (see also my **General Comment** on this point).

L368-375 – I have two comments about this paragraph. First, I believe this is a significant finding which could/should be better articulated both here and in the abstract and conclusion. By my understanding this places Vanderford Glacier as the 4th fastest retreating glacier in Antarctica (let alone East Antarctica) over the satellite era, only after Thwaites (0.8 kmyr = 3rd, as mentioned by the authors), Pine Island (0.9-1 kmyr = 2nd; Park et al., 2013; doi:10.1002/grl.50379) and Pope (3.3 kmyr = 1st; Millilo et al., 2022; doi:10.1038/s41561-021-00877-z).

That said, I was a little surprised to see no discussion of Rignot et al. (2019) here, who following Line 84 report the same rate of retreat between 1996 and 2017 (i.e. overlapping 90% of the present study's observational period). As such, I believe this paragraph (and elsewhere) should be reworked to better contextualize the author's findings against this study.

[NB: I don't think the above suggestion will necessarily detract from the novelty of the present study, as long as it's made clear your observations show the *continued* rapid retreat of Vanderford Glacier to 2020].

L416-419 - So what caused this significant GL retreat then? Discussion of passive ice according to Furst et al. is good, but strikes me as a convenient diversion away from what should be the main focus of this paragraph. A revised discussion of the expected key drivers of this phenomenon will make for a much more compelling read, even if the interpretation is 'the causes remain unknown and an important region for future research' or the like.

Fig.11 – If GLs are added to Fig.1 (see my comments on that figure above), then I'd consider moving Fig.11 to the supplementary information to avoid figure repetition in the main text.

Line 450-476. How has landfast sea ice changed between 1996 and present? An increasing amount of research has shown the importance of landfast sea ice/mélange for controlling calving rates in Antarctica (by congealing together / buttressing ice fronts), which I think should also be noted here.

(See papers by e.g. Greene et al. (doi:10.5194/tc-12-2869-2018), Francis et al. (doi:10.5194/tc-15-2147-2021), Fraser et al. (doi:10.5194/essd-12-2987-202), Arthur et al. (doi:10.1017/jog.2021.45), Christie et al. (doi:10.1038/s41561-022-00938-x) and Massom et al. (doi:10.1038/s41586-018-0212-1) for some recent examples of this in Antarctica).

L498-510 – OK, the majority of this paragraph goes on to discuss the importance of landfast sea ice for controlling terminus stability, but it seems largely out of place to me in this section. All things considered, to improve readability/clarity I would suggest merging Sections 4.2 and 4.3 into a new, single section which first overviews the observed glaciological changes (frontal retreat and thinning) and then discusses their possible links to sea-ice loss.

On a related note, recent work has also shown that decreased sea ice cover may also leave coastal regions vulnerable to the influence of storms or katabatic wind events which can disturb ocean surface slopes leading to enhanced strain-induced calving (see e.g. Francis et al. (doi:10.5194/tc-15-2147-2021; 10.1029/2021JD036424) and Christie et al. (doi:10.1038/s41561-022-00938-x) for recently documented examples of this phenomenon).

To my mind, these sorts of atmosphere-sea ice interactions (sea-ice debuttressing, wind-induced strain) are more likely to explain the observed patterns of calving than the mechanism proposed by Miles et al. 2016 (at least over the relatively short timescales considered here), because in Antarctica the majority of basal melting is confined to the GL (e.g. Rignot et al., 2013; doi:10.1126/science.1235798). This phenomenon doesn't occur by coincidence, and is because mCDW resides at the same depth as the GL. Elsewhere (including at the relatively much more shallow ice-shelf fronts), mCDW does not physically interact with the ice in such a way that can drive rapid frontal fracture/calving, and thus the more moderate thinning signals observed at those locations instead reflect lagged responses resulting from historical perturbations at the GL.

Section 4.4 – While interesting, most of this section reads like background information and less pertinent results which I'm not sure is best placed in (or integral to) the discussion. I suggest making this a supplementary discussion and alluding to it somewhere in Section 3.2.

Section 4.5 – Similar to Section 4.4, much of this section contains information which should belong in the Methods (especially given my comments on Lines 184-197 and 227). Following my suggestion for Line 340, I also think the discussion contained on Lines 539-545 could be removed as the signal-to-noise of break-in-slope-derived change relative to the 1996 DInSAR GL pick seems compelling. I further suspect that the pattern of retreat revealed by plotting each GL onto Fig.9 will support this conclusion.

Section 5 – While acceptable as written, the conclusion is very long and could/should be overhauled to provide a much more succinct, punchy summary of the key findings and implications only. I expect this could be done in half a page or less. Structure-wise, I also find the discussion going from *GL retreat* to *terminus change* and then back to *GL retreat* to be confusing, so suggest that the ideas contained in the paragraphs beginning Lines 547 and 573 could be merged into one coherent narrative.

L550 – Same comment as Lines 368-375.

L585 – In these days of reproducibility I would strongly encourage the authors to archive and openly share their terminus positions via a repository such as Cryoportal.

Table S1 – Table Caption – For completeness, insert 'optical' after 'Details of'.

**Technical Comments**

L33 – WAIS in this instance is a pronoun and so should not be preceded by 'the'. Similarly, 'embayment' should be capitalized. Multiple such blunders exist throughout the manuscript, so it will be worth carefully going through the text and weeding these out.

L37 – 'grounding line retreat' should read 'grounding-line retreat'. This should be changed throughout the manuscript, including in the title. Same for phrases such as 'ice-shelf thinning', 'ice-surface velocity', 'sea-ice production' etc.

L208 – Tense issue. Suggest rewording to: '…has an overall, associated error of +/- 100m'.

L221 – Grammar. Why does the use of MODIS imagery mean that the error is 250 m? Suggest rephrasing for improved clarity.

L291 – Typo? Think it should read '… at box IN accelerating from …'.

L323 – 'These trends'. What trends? Grammar. Perhaps the opening sentence of this paragraph could read something like: "Mirroring the elevation patterns observed by Schroder et al. (2019) and Nilsson et al. (2021), similar trends are found in the ICESat/ICESat-2-derived data of Smith et al. (2020) (Figure 7)".

L406 – Insert comma after '(Figure 6a)'.

L560 – Missing citation?

Section 7 – Text missing?

--- END ---

---

## Author Comment (AC1)

**Review 1 - Wei Ji Leong**

Hannah Picton

March 2023

**1 General Comments**

This manuscript presents an observational study of six outlet glaciers draining into Vincennes Bay in East Antarctica using products derived from optical satellite imagery, Synthetic Aperture Radar, and laser/radar altimetry sensors. It is an impressive piece of work that uses over half a century of remote sensing data from 1963 to 2022, relying on methods like grounding line delineation using manual digitisation and DInSAR, automated feature tracking for measuring ice velocity, correcting ice surface elevation trends over different time periods, and integrating all of that to investigate decadal scale trends. Overall, the manuscript is well written, with clear references to datasets and other relevant studies, and an interesting discussion about potential forcing mechanisms for the grounding line retreat while highlighting the continued need for DInSAR based grounding line mapping into the future.

That said, I do want to offer some suggestions on ways to improve the manuscript for publication. One of the major things that stood out was the emphasis on how basal melt from warm modified Circumpolar Deep Water (mCDW) is linked to the grounding line retreat at Vanderford Glacier, with a lower emphasis on other factors like changing atmospheric forcings (surface air temperature) and/or sea ice mélange conditions. The authors have been fairly careful in their wording in the Discussion section on this, and offer a convincing line of reasoning on how increased basal melt could result in the observed rapid rate of grounding line retreat, but it does feel like there is a sudden jump between the methods used and that part of the discussion, which could be resolved with a few more sentences on how basal melt rate could be estimated to confirm this line of reasoning. Other than that, there are only some minor points of clarification that could help to reduce some ambiguity, which I will highlight in the specific and technical comments section below.

We thank Wei Ji Leong for their positive feedback on the manuscript and constructive suggestions. Please find our responses to the specific and technical comments outlined below.

**2 Specific Comments**

**2.1 Data Availability**

For reproducibility purposes, please upload a copy of your digitised central flowlines, sampling boxes and terminus positions (shown in Figure 1a and described in Section 2.2 and 2.3) in a standard OGR format to a suitable data repository, so that others can benefit from this work too. Ideally, the surface elevation anomaly data (shown in Figure 6 and Supplementary Figure 2) should also be uploaded. If the data are not publicly accessible, a detailed explanation of why this is the case is required. I acknowledge the hard work and time spent manually digitising the lines, and would really appreciate this data to be shared with the Cryosphere community to save time on doing duplicate work.

The digitised central flowlines, sampling boxes and terminus positions will be uploaded to a suitable data repository in order to make the data publicly accessible. We will cite the exact data repository in the final version, should the manuscript be accepted. The surface elevation anomaly data will not be uploaded to a data repository, as each of the surface elevation change datasets used are already publicly available. Direct links to these open-access datasets are provided in 'Section 6. Data

Availability'. Details of how the surface elevation anomalies shown in Figure 6 and Supplementary Figure 2 were calculated are provided in the methods 'Section 2.4. Ice Surface Elevation'.

**2.2 Methods**

pg9, L229-230: Just to clarify, it states here that grounding line position changes was measured along the central flowline. However, on pg10, L257-258 Figure 2 caption, it says width-averaged terminus position changes, and at pg12, L273 Table 3, it just says terminus position change without mentioning if the change is measured just along the central flowline or width-averaged. For each of these instances, could you be explicit and mention what method was used to avoid any ambiguity?

Terminus position change was assessed using the box method (Moon and Joughin, 2008) and therefore represents a width-averaged value, whilst grounding line position change was simply measured along the central flowline. Table 3 and the associated caption will therefore be updated to explicitly state 'rate of width-averaged terminus position change'.

pg15, L318: Figure 6. This is a nice time-series plot. Two minor comments though. 1) This figure appears to be duplicated in Supplementary Figure 2, albeit with a different colour scheme? 2) Is there a reason for leaving out the Smith et al. (2020) ice surface elevation trends from the plot? At pg 7, L180-183, you mentioned doing some work to allow cross-comparison of the Schröder et al. (2019), Smith et al. (2020) and Nilsson et al. (2022) datasets, but aside from a brief comment on pg 15, L322 that the trends are similar, the Smith et al. (2020) ice surface elevation anomaly trend is not included in the plot? Would the plot look too confusing with a third dataset added in?

1) Figure 6 shows the monthly surface elevation change anomalies calculated within the GL box of each glacier, whilst Supplementary Figure 2 shows the monthly surface elevation change anomalies calculated within the IN box of each glacier. As stated on L316, the surface elevation change anomalies were observed to be very similar across the GL and IN boxes, hence the decision to only include one within the manuscript and show the other within the supplementary material.

2) The Smith et al. (2020) dataset simply provides a value representing the average rate of surface elevation change observed between 2003 and 2019. In contrast, each of the Schröder et al. (2019) and Nilsson et al. (2022) datasets provides monthly surface elevation change values. Whilst the Smith et al. (2020) dataset could be added to the plot in the form of a trendline with a constant gradient, an arbitrary elevation would have to be chosen in 2003 which may potentially be misleading. Inclusion of such a linear trendline could also distract from the interannual variability captured by the 24-month rolling means shown for the Schröder et al. (2019) and Nilsson et al. (2022) datasets. The decision was hence made to not include the Smith et al. (2020) dataset within Figure 6. Instead, the Smith et al. (2020) dataset was primarily used to validate and provide further certainty to the long-term trends extracted from the Schröder et al. (2019) and Nilsson et al. (2022) datasets. We will add some additional text to explain this point in Section 3.3 of the manuscript.

**2.3 Results**

pg12, L265: Just need some clarification on how the median rate of terminus position change over 1973-1991 for the 6 glaciers is calculated. Are you 1) taking the median rate from 1973-1991 for each glacier, and then taking the median of those values over 6 glaciers; or 2) taking the median rate per year over 6 glaciers, and then taking the median over the 1973-1991. In other words, are you taking the median time-wise then glacier-wise, or glacier-wise then time-wise?

Each glacier had a digitised terminus position in 1973 and 1991. The difference between these two positions was used to calculate the average rate of retreat in m/yr for each glacier between 1973 and 1991. The mean, median and standard deviation of these 6 average retreat rate values was then presented in Table 3. L264 - L269 can be edited, as required.

pg12, L273: Table 3. Could you please provide the raw time-series data for each of the 6 glaciers' terminus position for every year, either as a CSV table or in the supplementary file? This would help with the ambiguity mentioned above, and also it would be good too for future scientists to compare rates of change for individual glaciers over different time periods.

The raw terminus position time-series data will be provided in a supplementary figure.

**2.4   Discussion**

pg 19-22: One concern on the disconnect between the Methods section (which has not explicitly measured or modeled basal melt directly) and this Discussion section 4.1 (which details how mCDW enhances ice shelf thinning and leads to grounding line retreat). At the very least, there would be some mention of how such basal melt rates could be measured directly using radio-echo sounding, or estimated using changes in ice volume (using changes in ice surface elevation over an area and making some assumptions like hydrostatic equilibrium). This could be mentioned as 'Future Work'.

On lines 84-85, we provide a previous estimate made by Rignot et al. (2013) regarding the area-averaged basal melt rates across the Vincennes Bay ice shelves. We therefore don't believe this necessitates an additional 'future work' section, however we do agree that the manuscript would be improved by addressing the disconnection between the methods section and the discussion section, as suggested. We will therefore add a few more sentences to the Vanderford Glacier discussion (Section 4.1), emphasising that the accurate quantification of basal melt rates across the Vanderford Ice Shelf could further our understanding of the rapid grounding line retreat observed. We will then suggest how this could be quantified, with discussion of direct radio-echo sounding methods and use of estimated changes in ice volume, outlined.

**3   Technical Corrections**

**3.1   Abstract**

pg1, L13: "satellite aperture radar" - "synthetic aperture radar"

This will be changed to state 'synthetic aperture radar'.

**3.2   Introduction**

pg2, L33: "Recent mass loss". Please state a general time period. E.g. 2010s to 2020s, to be clearer about when this mass loss is happening.

This will be changed to state 'Recent mass loss from the WAIS (2000s to 2010s) has largely been concentrated within...'. The Feldmann & Levermann (2015) reference will therefore be removed.

pg2, L36-40: "The ice flow acceleration ... has been attributed to ice-shelf thinning and reduced buttressing, a process forced by the wind-driven intrusion of warm mCDW ...". This is a nice information-rich sentence, it might be good to mention 'basal ice-shelf melting' somewhere to be explicit that the forcing is from the bottom-up and not top-down.

This sentence will be changed to explicitly state that the ice-shelf thinning is driven by basal ice-shelf melting.

pg2, L52: "-57.0 ± 2 m" - "-57.0 ± 2 Gt/y

The units will be changed from m to Gt/y as suggested.

pg2, L56-58: "..., Miles et al. (2016) have observed ... 74% of Wilkes Land outlet glaciers measured to retreat between 2000 and 2012". Table 1 from Miles et al 2016 actually mentioned that the results from Wilkes Land (DB13) were obtained from a previous study, see Supplementary Table 6 by Miles et al 2013 (https://doi.org/10.1038/nature12382). In the 2013 paper, the date range is 2000-2010, while the 2016 uses 2000-2012, my interpretation is that the former (2000-2010) is the correct date range. It is recommended to cite the earlier 2013 paper instead of the 2016 paper as the canonical data source for this statistic. Also, 74% of XXX glaciers can be somewhat ambiguous (though it is used like so by Miles et al. 2016), I'd recommend stating the absolute number of glaciers that have retreated (n=39) in addition to the relative percentage. Overall, this sentence could be modified into something like "..., Miles et al. (2013) have observed widespread terminus retreat across the region, with 74% (n=39) outlet glaciers measured to retreat at a median rate of -63.6m/a-1 between 2000 and 2010".

This is a valid point and the suggested modified sentence will be included.

pg3, L71: Figure 1. Missing Longitude/Latitude gridlines or Polar Stereographic coordinate tick marks. Need to have some spatial coordinate reference system to set the geographical context of this area.

Tick marks and longitude/latitude gridlines will be added to Figure 1.

**3.3  Methods**

pg9, L242: Could you be a bit more specific about the REMA product used and version? Assuming that you are using REMA v1, and the mosaic instead of the strip DEMs? If using the mosaic, what spatial resolution, 10m, 100m, etc?

Surface topography profiles were extracted using the REMA v1 mosaic. Figure 9 in Howat et al. (2019) shows the spatial resolution of this dataset was between 2 and 8 m within the Vincennes Bay study area. These specific details will be clarified in Section 2.6 of the manuscript. It is important to note, however, that surface elevation was sampled at the same 500 m interval spacing as the bed elevation, as stated on L243.

**3.4  Results**

pg11, L258: Figure 3. If possible, provide longitude/latitude or polar stereographic coordinate tick marks.

Tick marks and longitude/latitude gridlines will be added to Figure 3.

pg11, L263: Figure 3 caption. Please clarify source of background optical satellite imagery, is it Sentinel-2, Landsat, or other?

The figure caption will be updated to state all background optical satellite imagery shown in Figure 3 was sourced from Sentinel-2B.

pg13, L282: Figure 4. Maybe better to replace 'Distance from VA (km)' with something like 'Distance from top of flowline (km)' or something like that?

Figure 4 x-axis caption will be changed to state 'Distance along VA flowline (km)'. The figure caption will also be edited to explicitly state that distance is measured in the downstream direction, with 0 km therefore representing the inland start point of each flowline.

pg16, L329: Figure 7 colorbar. The colour bins have only one label placed at the middle of the bins, e.g. the white box for +0.00. So does white represent -0.025 to +0.025, or +0.00 to +0.05? Could the numbers be shifted to make it less ambiguous?

The symbology used for Figure 7 was a linear interpolation between a maximum value of +0.10 m/yr and a minimum value of -0.20 m/yr. The labels shown on the colorbar therefore correspond to the exact value for the specific colour shown. For example, the white coloured bin represents an exact value of 0.00 m/yr, as labelled. However, although the colorbar is displayed in discrete bins, it is important to note that a continuous colorscheme is used within the figure. We think that this colorscheme is intuitive to the reader, with white colours indicative of little change, more intense blues indicative of increased rates of thinning, and more intense reds indicative of increased rates of thickening. Whilst this symbology could have been represented using a continuous colorbar with the minimum and maximum values labelled, we believe the binned examples provide more detail of the graduations to the reader. As the labels correspond to the specific colour bins, shifting the numbers would therefore be innapropriate.

We would also like to emphasise that the intention of Figure 7 is not for the reader to necessarily extract specific rates of surface elevation change, as this is primarily conducted using the monthly SEC datasets provided by Schröder et al. (2019) and Nilsson et al. (2022). Instead, the purpose of Figure 7 is to allow the reader to simply assess the spatial variation in ice surface elevation change observed within Vincennes Bay, which we believe is achieved.

pg18, L347: Figure 9. Maybe better to replace 'Distance from VA (km)' with something like 'Distance from top of flowline (km)' or something like that?

Figure 9 x-axis caption will be changed to state 'Distance along VA flowline (km)'. As explained previously, the figure caption will also be edited to explicitly state that distance is measured in the downstream direction, with 0 km therefore representing the inland start point of each flowline.

**3.5 Discussion**

pg22, L428: Figure 11. If possible, provide longitude/latitude or polar stereographic coordinate tick marks.

Tick marks and longitude/latitude gridlines will be added to Figure 11.

**3.6 Supplementary**

pg4, L17: Supplementary Figure 1. Missing Longitude/Latitude gridlines or Polar Stereographic coordinate tick marks. Need to have some spatial coordinate reference system to set the geographical context of this area.

Tick marks and longitude/latitude gridlines will be added to Supplementary Figure 1.

---

## Author Comment (AC2)

**Review 2 - Anonymous**

Hannah Picton

March 2023

In this manuscript, Picton et al. use a range of remotely sensed datasets to examine recent glaciological changes within Vincennes Bay, East Antarctica. Amongst other interesting indicators of glacier change, they report, most notably, upon the rapid retreat of Vanderford Glacier's grounding line at a rate of 0.8 km/yr between 1996 and 2020.

Overall, the paper is generally well-presented, well-structured and is scientifically robust, and even considering the observations of Vanderford Glacier's behavior alone – which places it as the 4th fastest retreating glacier in Antarctica over the satellite era – I believe this manuscript will be of broad interest to the readership of The Cryosphere. For this reason, I recommend publication. Prior to publication, however, I believe the manuscript has several limitations in its current form which should be addressed. These limitations are detailed in my comments below.

*We thank the anonymous reviewer for their positive feedback on the manuscript and constructive suggestions. Please find our responses to the general, minor and technical comments outlined below.*

**1 General Comments**

*Vanderford grounding-line retreat.* While the retreat rate reported here is undoubtedly significant, the problem is that it is for the most part not a new finding. This is because, as the authors themselves allude to on Line 84, Rignot et al. (2019) have previously reported upon this behavior as observed between 1996 and 2017 (over which time they also find a retreat rate of 0.8 km/yr). In this regard there are two key issues with the manuscript in its current form:

1) The grounding-line retreat-related findings – as presently reported at least – are perhaps more incremental than the narrative of the manuscript would suggest.

2) Apart from the introduction (Line 84), no further acknowledgement of Rignot et al.'s earlier observations is included, which could be misconstrued by some as slightly disingenuous.

To remedy these issues, I would suggest the authors:

1) Rework the text to contextualize their findings more clearly alongside this earlier research. (In e.g. the abstract and conclusion, phrasing like "Our results confirm extensive grounding-line retreat..." is used, but this doesn't make it explicit that this is a confirmation of a previously documented observation). More explicit follow-up discussion of the fact that the author's observations show continued retreat since 2017 would also be beneficial, and serve to demonstrate that their research goes beyond that discussed by Rignot et al.

*We appreciate that this could be misconstrued by some as slightly disingenuous, which was certainly not our intention. We will therefore ensure that is emphasised throughout the manuscript that our study not only supports the 0.8 km/yr of grounding line retreat **previously** observed at Vanderford by Rignot et al. (2019), but importantly also suggests **continued** rapid retreat has occurred between 2017 and 2020.*

2) What's also new (and arguably much more interesting) relative to the simple trend of 0.8 km/yr reported here and in Rignot et al. (2019) is the seemingly step-wise, temporally variable patterns of retreat observed between 1996 and 2021 (Fig. 8). I think a more explicit/nuanced discussion of this phenomenon and its links with e.g. changes in bed topography/MISI (or otherwise) as seen in Fig.

9 would make for a much more interesting read, while again going beyond that described in Rignot et al. (2019). (See also my **Minor Comment** on Fig.9 below).

Please see our comments below for further detail regarding Figure 8 and Figure 9. However, we will include more explicit discussion of the temporally variable patterns of retreat observed, particularly emphasising the sustained and consistent retreat observed at Vanderford Glacier.

*Structure/presentation of manuscript.* While generally well-presented/written overall, I believe the manuscript could also be overhauled in places (abstract, discussion and conclusion especially) to offer a more succinct / 'to the point' discussion of the key points and novel findings only. Perhaps most importantly, I think the structure of the discussion requires some careful refocusing, as at present it contains a lot of unnecessary details which should either be moved to the Methods, Results or Supplementary Information. Elements of the Discussion and Conclusion also have the tendency to jump back and forth between ideas and/or from one sub-section to the next, which I think should be corrected for improved readability. (See my **Minor Comments** below for some examples of these sorts of issues).

We recognise that parts of the manuscript, particularly the abstract, discussion and conclusion, could be made more succinct and corrected for improved readability. We will therefore make a number of changes, with specific examples provided in our response to the minor comments outlined below.

**2 Minor Comments**

L13 - The ice surface velocity, thinning and GL position datasets are also derived from remotely sensed techniques, so I suggest rephrasing the sentence to better convey this point.

This sentence will be rephrased in order to better convey this point.

L13 - synthetic aperture radar.

This will be changed to state 'synthetic aperture radar'.

L34 - driven - dominated (I agree that these are the two main glaciers, but other parts of the coast are just as sensitive to oceanic influence and are now contributing to these trends too).

'driven' will be changed to 'dominated'.

L57 - 'measured to' - 'having undergone'?

'measured to' will be changed to 'having undergone'.

L60 - Weaker easterly winds relative to what and where? This was unclear to me. This line should be revised to clarify this.

This will be clarified by stating 'Wilkes Land is characterised by a 'warm shelf' regime, whereby weak easterly winds and an absence of dense water formation facilitates the intrusion of warm CDW onto the continental shelf (Thompson et al., 2018; Stokes et al., 2022).'

Figure 1 - Nice figure. Could 1b be underlain by hillshade and/or contours to make the topography 'pop out' more? In 1a, I also strongly recommend displaying the most recent GL (or ideally all of those included in your timeseries if the figure doesn't look too cluttered) to give the reader an instant sense of how each glacier has retreated through time. To help the reader find the locations referred to in the text (e.g., L81), please also add lat/lon graticules to both panels (and all other maps for that matter).

Figure 1b will be underlain by a hillshade and/or contours in order to make the topography more clearly visible. Tick marks and longitude/latitude gridlines will also be added to all maps. However, the decision was made to only display the oldest GL position for two main reasons:

1) Figure 1 is primarily intended to introduce the regional context of the study area, particularly highlighting the location of Vincennes Bay relative to Totten Glacier and Law Dome, as well as displaying the positions of the central glacier flowlines and FT, GL and IN sampling boxes used for

analysis. We believe that the inclusion of additional GL positions from the timeseries detracts from these important features, with Figure 1 appearing rather cluttered.

2) Showing the most recent GL position would effectively display the main results of the paper within the first figure, which we don't think would necessarily be appropriate. We would also like to emphasise that for Vanderford, Adams and Anzac, the most recent GL position would be the 2020 AIS CCI position, whilst for Bond East, Bond West and Underwood it would be the 2014 MOA position. Displaying these different datasets prior to their introduction, description and explanation in Section 2.5, may be rather confusing to the reader.

L101+ - Here, I suggest removing the methodological detail behind the analyses performed (terminus position, velocity, elevation, GL change) as this information is better placed in the following section.

Whilst we recognise that this paragraph includes some methodological detail, these details are summarised very succinctly. We think that the inclusion of this short paragraph provides an important link between the introduction and the methods section, outlining how the stated aim is to be achieved. We therefore suggest that this section improves the overall readability of the manuscript and should remain unchanged.

L109 - 'USGS Earth Explorer data repository, with...'

'data repository' will be added.

L112 - – Why was co-registration of the earlier scenes performed? I presume this pertains to the old ephemerides and DEMs used for geocoding Landsat 1-4 images, and thus the need to ensure spatially consistent imaging through time? Worth stating that here if so. Also, why was co-registration not performed for the ARGON mosaic? Considerable positional errors can exist in those images, and likely exist in the Kim et al. mosaic too.

The Landsat 1-4 image scenes were co-registered due to geolocation issues and the need to ensure spatially consistent imaging through time, as suggested above. This will be clarified within the manuscript. In contrast, co-registration was not conducted for the ARGON imagery as the mosaic produced by Kim et al. (2007) has already been orthorectified using GCPs. Nonetheless, the geolocation accuracy was manually checked before any subsequent analysis was conducted; the positions of coastal rock outcrops, nunataks and visibly stable ice features were observed to match those observed from the more recent Sentinel-2B scene. We will clarify that such prior co-registration and additional manual verification was conducted for the ARGON mosaic within Section 2.1.

L119 - ... are hereafter referred to as ...

Sentence structure will be changed to state 'are hereafter referred to as'.

L132 - ... the AutoRIFT feature tracking algorithm.

'auto-RIFT algorithms' will be changed to 'the AutoRIFT feature tracking algorithm'.

L135 - Typo? I believe ENVEO records date back to 2014?

This is correct and will be rephrased to provide clarification. Whilst the ENVEO velocity mosaics are available at a monthly resolution between 2014 and 2021, this sentence should state that only those between 2019 and 2021 were used wtihin this study.

L136 - Suggest rephrasing to say "... using a combination of coherent and incoherent feature tracking techniques". (following Nagler et al. (2021; doi:10.1109/igarss47720.2021.9553514) which should also be cited here).

This will be rephrased with the Nagler et al. (2021) citation also added.

L184-197 - As the authors are aware, the grounding line and hinge line are two different components of the grounding zone, the latter being a proxy for the former which cannot be detected from satellite imaging. This fact should be stated somewhere in this paragraph, if anything to make it clear that the two components haven't been conflated.

The difference between the grounding line and the hinge line will be stated within this paragraph. We note that on L223-224, we later state: "Whilst the AIS CCI and MEaSUREs products both represent the inner limit of tidal flexure and thus approximate the actual grounding line position (Fricker et al., 2009; Rignot et al., 2016)...", thereby implying that the hinge line is used as a proxy for the grounding line position. Nonetheless, we appreciate that the manuscript would be improved with more explicit definitions of the two components at the beginning of the grounding line section.

L206/7 - '...was generated using...as applied to ERS-1 and ERS-2 imagery...'.

This sentence explains that the MEaSUREs grounding line product was not only generated using the same DInSAR technique as that used to generate the AIS CCI dataset, but that both datasets also used ERS-1 and ERS-2 imagery collected in 1996. We therefore think that use of the word 'also' is preferable over use of the word 'as'. The sentence will thus be changed to read:

"The Making Earth Science Data Records for Use in Research Environments (MEaSUREs) grounding line product was generated using similar DInSAR techniques as previously described for the AIS CCI product, also applied to ERS-1 and ERS-2 imagery collected in 1996 (Rignot et al., 2016)."

L216 - Should read '...and most seaward, spatially continuous break-in-slope' (since multiple discontinuous breaks-in-slope can exist downstream over, for example, pinning points or ice rumples).

'spatially continuous break-in-slope' will be added.

L227 - Following my comments on L184-197, I think it's important to state here that it's difficult to compare break-in-slope and hinge line positions in a direct/like-for-like manner, because they're ultimately measuring two very different components of the grounding zone. Therefore, even given the standalone instrument/technique errors shown in Table 2, any changes in GL position identified from the two techniques should be interpreted with caution, with any further discussion restricted to those exhibiting pronounced retreat where we can be confident change represents a true signal.

This is a valid point and will be emphasised within Section 2.5.

L230-234 - I'm not sure I follow this, and specifically why the two GLs would be so far apart. Geocoding issues? Tidal effects (if different imaging dates)? Both? Additional information explaining this would be good here.

We agree that this is slightly confusing, and would like to emphasise that we were also puzzled as to why the two grounding lines would be so far apart. This motivated our decision to measure retreat relative to the most landward observed 1996 position, in order to ensure that we reported the most conservative estimate of grounding line retreat at Underwood Glacier. No explicit explanation is provided in the documentation associated with the dataset (Rignot et al., 2016), with the 'Quality Assessment' simply stating:

"The standard error is ±100 m, with greater geolocation variations locally. In some cases, large (km) short-term and long-term migrations are present. The quality of the grounding line mapping depends on the satellite data used, the length of the interferometric baseline (short baselines yield more accurate positioning), the amplitude of the differential tides, phase coherence (high phase coherence means less noise), and the frequency of revisits."

Whilst tidal effects were considered, the metadata provided with each GL states that both positions were mapped using imagery from the same European Space Agency Earth Remote Sensing Satellite (ERS), across the same acquisition dates (1996/2/6, 1996/2/7, 1996/3/12 and 1996/3/13). We will therefore try to contact the creators of the dataset in order to seek an explanation for the different GL positions. Any clarification provided will subsequently be added to the manuscript.

Fig.2 - Suggest noting the non-linear x-axis scale in the caption or, even better, editing the figures to show a linear scale.

The x-axis scale is linear, however the tick marks were placed at irregular spacings. This decision was made in an attempt to allow the reader to better understand the exact timings of the terminus digitisations, particularly across the earlier years of the record. For example, it can clearly be seen that the terminus positions of Vanderford, Adams and Anzac glacier were first observed in 1963,

whereas Bond East, Bond West and Underwood Glacier were first observed in 1973. However, we appreciate that such irregular spacing may be confusing to the reader and potentially misinterpreted as a non-linear scale. The labelled tick marks will therefore instead be placed at regular 10-year intervals between 1960 and 2020, with minor tick marks shown every 5 years.

Fig.3 - Nice figure. Caption should read '...from the USGS Earth Eplorer data repository' or similar.

'data repository' will be added.

Fig.4 - Nice figure, but what does, for example 'Distance from VA' mean on the x-axis? Does this literally mean from the 'VA' label on Figure 1? If so, suggest changing the notion to read 'Distance downstream' *or similar), and annotating VA to VA on both Fig.1 and the cross section profiles of Fig.4.

The Figure 4 x-axis title will be changed to state 'Distance along VA flowline (km)', 'Distance along AD flowline (km)', 'Distance along AN flowline (km)', etc. The figure caption will also be edited to explicitly state that distance is measured in the downstream direction, with 0 km therefore representing the inland start point of each flowline.

L290 - Associated errors. This is great, but please also show these errors on Fig 5 too.

We recognise that the associated errors could be added to Figure 5, but because each individual graph displays the mean annual velocity across each of the IN, GL and FT boxes, this addition makes Figure 5 rather busy. Instead, we believe it would be more helpful to maintain the clarity of Figure 5, but ensure that for each specific velocity value reported in Section 3.2, the associated error is stated. For example, L290 will be changed to state:

"Whilst ice surface velocity was seen to increase by 12% across the FT of Anzac Glacier between 2009 and 2021 (Figure 5c), this velocity increase was not deemed notable, with the absolute value of acceleration (30 m/yr) being smaller than the associated error (± 82 m/yr)."

Figs. 6 and 7 - These are clear, detail-rich figures. My one thought, however, is whether they could be merged somehow to save the reader flicking back and forth between figs?

The Smith et al. (2020) dataset shown in Figure 7 simply provides a value representing the average rate of surface elevation change observed between 2003 and 2019. In contrast, each of the Schröder et al. (2019) and Nilsson et al. (2022) datasets shown in Figure 6 represent monthly surface elevation change values. Whilst the Smith et al. (2020) dataset may be added to Figure 6 in the form of a linear trendline with a constant gradient, an arbitrary elevation would have to be chosen in 2003 which could potentially be misleading. Inclusion of such a linear trendline may also distract from the interannual variability captured by the 24-month rolling means calculated from the Schröder et al. (2019) and Nilsson et al. (2022) datasets. The decision was therefore made to not include the Smith et al. (2020) dataset within Figure 6. Instead, the Smith et al. (2020) dataset was primarily used to validate and provide further certainty to the long-term trends extracted from the Schröder et al. (2019) and Nilsson et al. (2022) datasets, with Figure 7 used to highlight the spatial variation in ice surface elevation change observed within Vincennes Bay.

L340 - Following my comment on L227, I think it's important to mention here that the retreat observed falls greatly outside sensor error limits, and likely also any between-sensor uncertainties associated with combined DInSAR/break-in-slope comparisons.

This is a helpful point that will be added.

Fig 8 - This is a nice figure, although I question whether it's integral to the interpretation presented. I think a revised version of Fig. 9 with each GL position shown would be far more impactful, with Fig. 8 moved to the supplementary information instead. Either way, remove the word 'manually' from the caption since the methods states that both break-in-slope and DInSAR mapping was carried out in this way.

We appreciate the reviewers comment, but think that Figure 8 is important and integral to the manuscript for two primary reasons:

1) As previously stated in the general comments, Figure 8 is useful for showing the temporally variable patterns of grounding line retreat observed between 1996 and 2020. It highlights that the grounding line positions mapped from optical imagery (ASAID and MOA datasets) were digitised in near-identical locations at each of the Vincennes Bay outlet glaciers, with the notable exception of Vanderford Glacier. Such near-identical locations would not be visible to the reader if placed as vertical lines on Figure 9, which we think would detract from the associated discussion.

2) The symbology used within Figure 8 allows the reader to assess the different methods used to quantify grounding line retreat. We believe this is unique to the manuscript and provides important context for the later discussion of the importance of accurate DInSAR grounding line mapping.

Nonetheless, the figure caption will be changed to remove the word 'manually' as suggested.

Fig 9 - Please add each GL location to portray the temporal evolution of retreat more clearly, and if needed revise the results/discussion to reflect any temporal patterns of retreat this reveals (see also my **General Comment** on this point).

As discussed above, we don't think that Figure 9 is suitable for displaying the temporal evolution of grounding line retreat clearly. This is primarily due to the near-identical ASAID and MOA grounding line positions observed at 5 of the 6 studied glaciers, which would be difficult for the reader to observe as dated vertical lines. However, Figure 9 still provides important context of both the bed and ice surface elevation along each flowline. We therefore think there is value in keeping both Figure 8 and Figure 9 in the manuscript, and note that the total of 12 figures is not considered excessive for a paper published in *The Cryosphere*. We will, however, include additional discussion of the temporal patterns of retreat observed using both figures, particularly emphasising the sustained and consistent retreat observed at Vanderford Glacier.

L368-375 - I have two comments about this paragraph. First, I believe this is a significant finding which could/should be better articulated both here and in the abstract and conclusion. By my understanding this places Vanderford Glacier as the 4[th] fastest retreating glacier in Antarctica (let alone East Antarctica) over the satellite era, only after Thwaites (0.8 kmyr = 3[rd], as mentioned by the authors, Pine Island (0.9-1 kmyr= 2[nd], Park et al., 2013; doi:10.1002/grl.50379) and Pope (3.3 kmyr = 1[st], Millilo et al., 2022; doi:10.1038/s41561-021-00877-z).

We will emphasise the significance of the rapid rate of grounding line retreat observed at Vanderford Glacier more, both in this paragraph and within the abstract and conclusion.

That said, I was a little surprised to see no discussion of Rignot et al. (2019) here, who following Line 84 report the same rate of retreat between 1996 and 2017 (i.e. overlapping 90% of the present study's observational period). As such, I believe this paragraph (and elsewhere) should be reworked to better contextualize the author's findings against this study.

[NB: I don't think the above suggestion will necessarily detract from the novelty of the present study, as long as it's made clear your observations show the *continued* rapid retreat of Vanderford Glacier to 2020].

As stated in our earlier response to the general comments, we appreciate that the manuscript needs to better emphasise that our study not only supports the 0.8 km/yr of grounding line retreat **previously** observed at Vanderford by Rignot et al. (2019), but importantly also suggests **continued** rapid retreat has occurred between 2017 and 2020.

However, we think it is important to emphasise that the Rignot et al. (2019) paper aimed to 'evaluate the state of the mass balance of the Antarctic Ice Sheet over the last four decades' and therefore placed less focus on specific individual glaciers. As a result, Vanderford Glacier was only mentioned twice in the entire manuscript, with a single sentence regarding the extensive grounding line retreat: 'Vanderfjord experienced a spectacular grounding line retreat of 17 km between 1996 and 2017 (SI Appendix Fig. S2)'. The associated supplementary figure simply shows double difference interferograms from 1996 and 2017, with the grounding line positions displayed.

Whilst this paragraph (and elsewhere) will be reworked to better contextualize our findings against that of Rignot et al. (2019), we believe our manuscript offers detailed insights unable to be covered within the much wider-scale analysis conducted by Rignot et al. (2019).

L416-419 - So what caused this significant GL retreat then? Discussion of passive ice according to Furst et al. is good, but strikes me as a convenient diversion away from what should be the main focus of this paragraph. A revised discussion of the expected key drivers of this phenomenon will make for a much more compelling read, even if the interpretation is 'the causes remain unknown and an important region for future research' or the like.

We think that Section 4.1 outlines what may be causing the significant GL retreat, with clear discussion of the potential intrusion of warm mCDW along the Vanderford Trench at depth. However, we will try to strengthen this discussion further by including content from a later suggestion made by the anonymous reviewer. We will emphasise that in Antarctica, the majority of basal melting is confined to the GL and thus, as stated below, the more moderate thinning signals observed elsewhere may instead reflect lagged responses resulting from historical perturbations at the GL.

Fig 11 - If GLs are added to Fig.1 (see my comments on that figure above), then I'd consider moving Fig.11 to the supplementary information to avoid figure repetition in the main text.

As explained previously, we think the addition of further GLs makes Figure 1 too cluttered and displays the main findings of the paper before the context has been established, potentially confusing the reader. We therefore think Figure 11 should remain in the manuscript and note that a total of 12 figures is not considered excessive for a paper published in *The Cryosphere*. It should also be noted that the Vanderford flowline is extended in Figure 11 and therefore serves an important role in highlighting the inland retrograde slope observed along the Vanderford Trench.

L450-476 - How has landfast sea ice changed between 1996 and present? An increasing amount of research has shown the importance of landfast sea ice/mélange for controlling calving rates in Antarctica (by congealing together / buttressing ice fronts), which I think should also be noted here. (See papers by e.g. Greene et al. (doi:10.5194/tc-12-2869-2018), Francis et al. (doi:10.5194/tc-15-2147-2021), Fraser et al. (doi:10.5194/essd-12-2987-202), Arthur et al.(doi:10.1017/jog.2021.45), Christie et al. (doi:10.1038/s41561-022-00938-x) and Massom et al. (doi:10.1038/s41586-018-0212-1) for some recent examples of this in Antarctica).

L498-510 – OK, the majority of this paragraph goes on to discuss the importance of landfast sea ice for controlling terminus stability, but it seems largely out of place to me in this section. All things considered, to improve readability/clarity I would suggest merging Sections 4.2 and 4.3 into a new, single section which first overviews the observed glaciological changes (frontal retreat and thinning) and then discusses their possible links to sea-ice loss.

On a related note, recent work has also shown that decreased sea ice cover may also leave coastal regions vulnerable to the influence of storms or katabatic wind events which can disturb ocean surface slopes leading to enhanced strain-induced calving (see e.g. Francis et al. (doi:10.5194/tc-15-2147- 2021; 10.1029/2021JD036424) and Christie et al. (doi:10.1038/s41561-022-00938-x) for recently documented examples of this phenomenon).

To my mind, these sorts of atmosphere-sea ice interactions (sea-ice debuttressing, wind-induced strain) are more likely to explain the observed patterns of calving than the mechanism proposed by Miles et al. 2016 (at least over the relatively short timescales considered here), because in Antarctica the majority of basal melting is confined to the GL (e.g. Rignot et al., 2013; doi:10.1126/science.1235798). This phenomenon doesn't occur by coincidence, and is because mCDW resides at the same depth as the GL. Elsewhere (including at the relatively much more shallow ice-shelf fronts), mCDW does not physically interact with the ice in such a way that can drive rapid frontal fracture/calving, and thus the more moderate thinning signals observed at those locations instead reflect lagged responses resulting from historical perturbations at the GL.

In response to the above four comments, we will consider merging Section 4.2 and 4.3 into a single section. However, we would like to highlight that whilst both sections discuss the same sea-ice forcing mechanism, they discuss glaciological changes on very different temporal scales. Section 4.2 analyses decadal patterns across the entire study period, whilst Section 4.3 focuses on the comparatively short and recent time period of 2017-2020. We therefore suggest that maintaining some kind of separation, even if simply just beginning a new paragraph, would be preferable. Nonetheless, we will strengthen this section by including additional discussion of the potential influence of decreased sea ice cover on the vulnerability of coastal regions to storms or katabatic wind events.

Section 4.4 – While interesting, most of this section reads like background information and less pertinent results which I'm not sure is best placed in (or integral to) the discussion. I suggest making this a supplementary discussion and alluding to it somewhere in Section 3.2.

We appreciate that this section is not integral to the main discussion of the paper. However, with Bond West Glacier observed to reach speeds in excess of Shirase Glacier, previously thought to be the fastest flowing outlet glacier in East Antarctica, we believe it will still be of interest to the readership of *The Cryosphere*. We will therefore include a shortened and more concise version of Section 4.4 within Section 3.2, simply presenting our findings as a noteworthy result, rather than providing a more lengthy discussion.

Section 4.5 – Similar to Section 4.4, much of this section contains information which should belong in the Methods (especially given my comments on Lines 184-197 and 227). Following my suggestion for Line 340, I also think the discussion contained on Lines 539-545 could be removed as the signal-to-noise of break-in-slope-derived change relative to the 1996 DInSAR GL pick seems compelling. I further suspect that the pattern of retreat revealed by plotting each GL onto Fig.9 will support this conclusion.

We think that Section 4.5 represents an important discussion point of the paper, summarising the main methodological finding of the study. This section therefore builds upon the findings of Figure 8, providing a unique assessment of the different methods used to quantify grounding line retreat. With this glaciological parameter representing such an important indicator of dynamic change, we believe this discussion will be of interest to the readership of *The Cryosphere*.

However, we agree that the signal-to-noise of break-in-slope derived change relative to the 1996 DInSAR pick seems compelling. We will therefore alter the tone of L539-545 to suggest that whilst inferences made using break-in-slope derived grounding line positions are less certain than those made using DInSAR derived grounding line positions, the compelling signal-to-noise ratio suggests that reduced rates of grounding line retreat have indeed occurred at Adams, Anzac, Bond East, Bond West and Underwood Glacier, in comparison to Vanderford Glacier.

Section 5 – While acceptable as written, the conclusion is very long and could/should be overhauled to provide a much more succinct, punchy summary of the key findings and implications only. I expect this could be done in half a page or less. Structure-wise, I also find the discussion going from *GL retreat* to *terminus change* and then back to *GL retreat* to be confusing, so suggest that the ideas contained in the paragraphs beginning Lines 547 and 573 could be merged into one coherent narrative.

We don't consider three paragraphs to be too excessive for a conclusion, however we will make the conclusion more succinct wherever possible. We also appreciate that switching from *GL retreat* to *terminus change* and then back to *GL retreat* may be confusing to the reader and will therefore restructure the conclusion to form one coherent narrative.

L550 – Same comment as Lines 368-375.

As stated in response to L368-375, we will emphasise the significance of the rapid rate of grounding line retreat observed at Vanderford Glacier more.

L585 – In these days of reproducibility I would strongly encourage the authors to archive and openly share their terminus positions via a repository such as Cryoportal.

The digitised central flowlines, sampling boxes and terminus positions will be uploaded to a suitable data repository in order to make the data publicly accessible. We will cite the exact data repository in the final version, should the manuscript be accepted.

Table S1 – Table Caption – For completeness, insert 'optical' after 'Details of'.

Caption will be changed to state 'Details of the optical satellite imagery used within this study'.

**3 Technical Comments**

L33 – WAIS in this instance is a pronoun and so should not be preceded by 'the'. Similarly,'embayment' should be capitalized. Multiple such blunders exist throughout the manuscript, so it will be worth carefully going through the text and weeding these out.

The sentence reads 'Recent mass loss from the WAIS has largely been concentrated...'. We therefore believe that in order to be grammatically correct, use of the word 'the' is required before 'WAIS'. We note that this convention is regularly adopted within the literature, including published papers within *The Cyrosphere*. Some recent examples include Maclennan et al. (2023) (https://doi.org/10.5194/tc-17-865-2023), Holland et al. (2022) (https://doi.org/10.5194/tc-16-5085-2022) and Schlemm et al. (2022) (https://doi.org/10.5194/tc-16-1979-2022).

However, we will ensure that 'embayment' is capitalised throughout the manuscript.

L37 – 'grounding line retreat' should read 'grounding-line retreat'. This should be changed throughout the manuscript, including in the title. Same for phrases such as 'ice-shelf thinning', 'ice-surface velocity', 'sea-ice production' etc.

'grounding line retreat' will be changed to 'grounding-line retreat' throughout the manuscript, including in the title. Such changes will also be completed for the suggested phrases outlined above.

L208 – Tense issue. Suggest rewording to: '... has an overall, associated error of +/- 100m'.

This will be reworded as suggested in order to correct the tense issue.

L221 – Grammar. Why does the use of MODIS imagery mean that the error is 250 m? Suggest rephrasing for improved clarity.

This sentence will be rephrased in order to improve the clarity.

L291 – Typo? Think it should read '... at box IN accelerating from ...'.

This will be changed to state '... at box IN accelerating from ...', as suggested.

L323 – 'These trends'. What trends? Grammar. Perhaps the opening sentence of this paragraph could read something like: "Mirroring the elevation patterns observed by Schroder et al. (2019) and Nilsson et al. (2021), similar trends are found in the ICESat/ICESat-2-derived data of Smith et al. (2020) (Figure 7)".

Sentence will be changed to that suggested above.

L406 – Insert comma after '(Figure 6a)'.

Comma will be inserted after '(Figure 6a)'.

L560 – Missing citation?

The prediction of both accelerated thinning and increased ice surface velocities represents our own suggestion based on the results and discussion presented, rather than that of another author.

Section 7 – Text missing?

A link to the supplementary material associated with this paper will be provided.

---

## Author Response (AR1)

**Authors Response**

**Hannah Picton**

**June 2023**
* * *
**The following section provides a point-by-point response to Reviewer 1, Wei Ji Leong. The reviewer's comments are shown in black, followed by our response shown in blue.**

**1 General Comments**

This manuscript presents an observational study of six outlet glaciers draining into Vincennes Bay in East Antarctica using products derived from optical satellite imagery, Synthetic Aperture Radar, and laser/radar altimetry sensors. It is an impressive piece of work that uses over half a century of remote sensing data from 1963 to 2022, relying on methods like grounding line delineation using manual digitisation and DInSAR, automated feature tracking for measuring ice velocity, correcting ice surface elevation trends over different time periods, and integrating all of that to investigate decadal scale trends. Overall, the manuscript is well written, with clear references to datasets and other relevant studies, and an interesting discussion about potential forcing mechanisms for the grounding line retreat while highlighting the continued need for DInSAR based grounding line mapping into the future.

That said, I do want to offer some suggestions on ways to improve the manuscript for publication. One of the major things that stood out was the emphasis on how basal melt from warm modified Circumpolar Deep Water (mCDW) is linked to the grounding line retreat at Vanderford Glacier, with a lower emphasis on other factors like changing atmospheric forcings (surface air temperature) and/or sea ice mélange conditions. The authors have been fairly careful in their wording in the Discussion section on this, and offer a convincing line of reasoning on how increased basal melt could result in the observed rapid rate of grounding line retreat, but it does feel like there is a sudden jump between the methods used and that part of the discussion, which could be resolved with a few more sentences on how basal melt rate could be estimated to confirm this line of reasoning. Other than that, there are only some minor points of clarification that could help to reduce some ambiguity, which I will highlight in the specific and technical comments section below.

We thank Wei Ji Leong for their positive feedback on the manuscript and constructive suggestions. Please find our responses to the specific and technical comments outlined below.

**2 Specific Comments**

**2.1 Data Availability**

For reproducibility purposes, please upload a copy of your digitised central flowlines, sampling boxes and terminus positions (shown in Figure 1a and described in Section 2.2 and 2.3) in a standard OGR format to a suitable data repository, so that others can benefit from this work too. Ideally, the surface elevation anomaly data (shown in Figure 6 and Supplementary Figure 2) should also be uploaded. If the data are not publicly accessible, a detailed explanation of why this is the case is required. I acknowledge the hard work and time spent manually digitising the lines, and would really appreciate this data to be shared with the Cryosphere community to save time on doing duplicate work.

The manually digitised central flowlines, sampling boxes and terminus positions have been uploaded to the UK Polar Data Centre and can be accessed at: https://doi.org/10.5285/4D4BD383-F6BB-4476-9058-D883B7706B26. This dataset is under embargo until paper publication, but can be accessed by the reviewers if required.

The surface elevation anomaly data has not been uploaded to a data repository, as each of the surface elevation change datasets used are already publicly available. Direct links to these open-access datasets are provided in 'Section 6. Data Availability'. Details of how the surface elevation anomalies shown in Figure 6 and Supplementary Figure 2 were calculated are provided in the methods 'Section 2.4. Ice Surface Elevation'.

**2.2 Methods**

pg9, L229-230: Just to clarify, it states here that grounding line position changes was measured along the central flowline. However, on pg10, L257-258 Figure 2 caption, it says width-averaged terminus position changes, and at pg12, L273 Table 3, it just says terminus position change without mentioning if the change is measured just along the central flowline or width-averaged. For each of these instances, could you be explicit and mention what method was used to avoid any ambiguity?

Terminus position change was assessed using the box method (Moon and Joughin, 2008) and therefore represents a width-averaged value, whilst grounding line position change was simply measured along the central flowline. Table 3 and the associated caption have been updated to explicitly state 'rate of width-averaged terminus position change'. The raw terminus position data has also been provided in Table S3 to provide further clarity.

pg15, L318: Figure 6. This is a nice time-series plot. Two minor comments though. 1) This figure appears to be duplicated in Supplementary Figure 2, albeit with a different colour scheme? 2) Is there a reason for leaving out the Smith et al. (2020) ice surface elevation trends from the plot? At pg 7, L180-183, you mentioned doing some work to allow cross-comparison of the Schröder et al. (2019), Smith et al. (2020) and Nilsson et al. (2022) datasets, but aside from a brief comment on pg 15, L322 that the trends are similar, the Smith et al. (2020) ice surface elevation anomaly trend is not included in the plot? Would the plot look too confusing with a third dataset added in?

1) Figure 6 shows the monthly surface elevation change anomalies calculated within the GL box of each glacier, whilst Supplementary Figure 2 shows the monthly surface elevation change anomalies calculated within the IN box of each glacier. As stated on L335-336 [of the revised manuscript], surface elevation change anomalies were observed to be very similar across the GL and IN boxes, hence the decision to only include one within the manuscript and show the other within the supplementary material.

2) The Smith et al. (2020) dataset simply provides a value representing the average rate of surface elevation change observed between 2003 and 2019. In contrast, each of the Schröder et al. (2019) and Nilsson et al. (2022) datasets provides monthly surface elevation change values. Whilst the Smith et al. (2020) dataset could have been added to the plot in the form of a trendline with a constant gradient, an arbitrary elevation would have to be chosen in 2003 which may potentially be misleading. Inclusion of such a linear trendline could also distract from the interannual variability captured by the 24-month rolling means shown for the Schröder et al. (2019) and Nilsson et al. (2022) datasets. The decision was therefore made to not include the Smith et al. (2020) dataset within Figure 6. Instead, the Smith et al. (2020) dataset was primarily used to validate and provide further certainty to the long-term trends extracted from the Schröder et al. (2019) and Nilsson et al. (2022) datasets.

**2.3 Results**

pg12, L265: Just need some clarification on how the median rate of terminus position change over 1973-1991 for the 6 glaciers is calculated. Are you 1) taking the median rate from 1973-1991 for each glacier, and then taking the median of those values over 6 glaciers; or 2) taking the median rate per year over 6 glaciers, and then taking the median over the 1973-1991. In other words, are you taking the median time-wise then glacier-wise, or glacier-wise then time-wise?

Each glacier had a digitised terminus position in 1973 and 1991. The difference between these two

positions was used to calculate the average rate of retreat in m/yr for each glacier between 1973 and 1991. The mean, median and standard deviation of these 6 average retreat rate values was then presented in Table 3. We have provided the raw terminus position data in Table S3 to provide further clarity regarding these calculations.

pg12, L273: Table 3. Could you please provide the raw time-series data for each of the 6 glaciers' terminus position for every year, either as a CSV table or in the supplementary file? This would help with the ambiguity mentioned above, and also it would be good too for future scientists to compare rates of change for individual glaciers over different time periods.

The raw terminus position time-series data has been provided in Table S3.

**2.4 Discussion**

pg 19-22: One concern on the disconnect between the Methods section (which has not explicitly measured or modeled basal melt directly) and this Discussion section 4.1 (which details how mCDW enhances ice shelf thinning and leads to grounding line retreat). At the very least, there would be some mention of how such basal melt rates could be measured directly using radio-echo sounding, or estimated using changes in ice volume (using changes in ice surface elevation over an area and making some assumptions like hydrostatic equilibrium). This could be mentioned as 'Future Work'.

On L82-83 and L445 [of the revised manuscript], we provide a previous estimate made by Rignot et al. (2013) regarding the area-averaged basal melt rates across the Vincennes Bay ice shelves. We therefore didn't believe this necessitated an additional 'future work' section, but agree that the manuscript would be improved by addressing the disconnection between the methods section and the discussion section, as suggested. We have therefore added a short paragraph to the Vanderford Glacier discussion (Section 4.1 - Lines 444-453), emphasising that the accurate quantification of basal melt rates across the Vanderford Ice Shelf could further our understanding of the rapid grounding line retreat observed. We outline both indirect (volume-flux divergence) and direct (radio-echo sounding) methods that could be employed.

**3 Technical Corrections**

**3.1 Abstract**

pg1, L13: "satellite aperture radar" - "synthetic aperture radar"

This has been changed to state 'synthetic aperture radar'.

**3.2 Introduction**

pg2, L33: "Recent mass loss". Please state a general time period. E.g. 2010s to 2020s, to be clearer about when this mass loss is happening.

This has been changed to state 'Recent mass loss from the WAIS (2000s to 2010s) has largely been concentrated within...'. The Feldmann & Levermann (2015) reference has therefore been removed.

pg2, L36-40: "The ice flow acceleration ... has been attributed to ice-shelf thinning and reduced buttressing, a process forced by the wind-driven intrusion of warm mCDW ...". This is a nice information-rich sentence, it might be good to mention 'basal ice-shelf melting' somewhere to be explicit that the forcing is from the bottom-up and not top-down.

This sentence has been split in two in order to explicitly state that the ice-shelf thinning is driven by basal ice-shelf melting.

pg2, L52: "-57.0 ± 2 m" - "-57.0 ± 2 Gt/y

The units have been changed from m to Gt/y as suggested.

pg2, L56-58: "..., Miles et al. (2016) have observed ... 74% of Wilkes Land outlet glaciers measured to retreat between 2000 and 2012". Table 1 from Miles et al 2016 actually mentioned that the results

from Wilkes Land (DB13) were obtained from a previous study, see Supplementary Table 6 by Miles et al 2013 (https://doi.org/10.1038/nature12382). In the 2013 paper, the date range is 2000-2010, while the 2016 uses 2000-2012, my interpretation is that the former (2000-2010) is the correct date range. It is recommended to cite the earlier 2013 paper instead of the 2016 paper as the canonical data source for this statistic. Also, 74% of XXX glaciers can be somewhat ambiguous (though it is used like so by Miles et al. 2016), I'd recommend stating the absolute number of glaciers that have retreated (n=39) in addition to the relative percentage. Overall, this sentence could be modified into something like "..., Miles et al. (2013) have observed widespread terminus retreat across the region, with 74% (n=39) outlet glaciers measured to retreat at a median rate of -63.6m/a-1 between 2000 and 2010".

The suggested modified sentence has been included.

pg3, L71: Figure 1. Missing Longitude/Latitude gridlines or Polar Stereographic coordinate tick marks. Need to have some spatial coordinate reference system to set the geographical context of this area.

Tick marks and longitude/latitude gridlines have been added to Figure 1.

**3.3  Methods**

pg9, L242: Could you be a bit more specific about the REMA product used and version? Assuming that you are using REMA v1, and the mosaic instead of the strip DEMs? If using the mosaic, what spatial resolution, 10m, 100m, etc?

This section has been modified to state that surface topography profiles were extracted using the REMA v1 mosaic, which has a spatial resolution between 2 and 8 m within the study area.

**3.4  Results**

pg11, L258: Figure 3. If possible, provide longitude/latitude or polar stereographic coordinate tick marks.

Tick marks and longitude/latitude gridlines have been added to Figure 3.

pg11, L263: Figure 3 caption. Please clarify source of background optical satellite imagery, is it Sentinel-2, Landsat, or other?

The figure caption has been updated to state all background optical satellite imagery shown in Figure 3 was sourced from Sentinel-2B.

pg13, L282: Figure 4. Maybe better to replace 'Distance from VA (km)' with something like 'Distance from top of flowline (km)' or something like that?

The Figure 4 x-axis caption has been changed to state 'Distance along VA flowline (km)'. The figure caption has also been edited to explicitly state that distance is measured in the along-flow direction, with 0 km representing the inland start point of each respective flowline.

pg16, L329: Figure 7 colorbar. The colour bins have only one label placed at the middle of the bins, e.g. the white box for +0.00. So does white represent -0.025 to +0.025, or +0.00 to +0.05? Could the numbers be shifted to make it less ambiguous?

The symbology used for Figure 7 was a linear interpolation between a maximum value of +0.10 m/yr and a minimum value of -0.20 m/yr. The labels shown on the colorbar therefore correspond to the exact value for the specific colour shown. For example, the white coloured bin represents an exact value of 0.00 m/yr, as labelled. However, although the colorbar is displayed in discrete bins, it is important to note that a continuous colorscheme is used within the figure. We think that this colorscheme is intuitive to the reader, with white colours indicative of little change, more intense blues indicative of increased rates of thinning, and more intense reds indicative of increased rates of thickening. Whilst this symbology could have been represented using a continuous colorbar with the minimum and maximum values labelled, we believe the binned examples provide more detail of the

graduations to the reader. As the labels correspond to the specific colour bins, shifting the numbers would therefore be innapropriate.

We would also like to emphasise that the intention of Figure 7 is not for the reader to necessarily extract specific rates of surface elevation change, as this is primarily conducted using the monthly SEC datasets provided by Schröder et al. (2019) and Nilsson et al. (2022). Instead, the purpose of Figure 7 is to allow the reader to simply assess the spatial variation in ice surface elevation change observed within Vincennes Bay, which we believe is achieved.

pg18, L347: Figure 9. Maybe better to replace 'Distance from VA (km)' with something like 'Distance from top of flowline (km)' or something like that?

Figure 9 x-axis caption has been changed to state 'Distance along VA flowline (km)'. As explained previously, the figure caption has also been edited to explicitly state that distance is measured in the along-flow direction, with 0 km representing the inland start point of each respective flowline.

**3.5  Discussion**

pg22, L428: Figure 11. If possible, provide longitude/latitude or polar stereographic coordinate tick marks.

Tick marks and longitude/latitude gridlines have been added to Figure 11.

**3.6  Supplementary**

pg4, L17: Supplementary Figure 1. Missing Longitude/Latitude gridlines or Polar Stereographic coordinate tick marks. Need to have some spatial coordinate reference system to set the geographical context of this area.

Tick marks and longitude/latitude gridlines have been added to Supplementary Figure 1.

**The following section provides a point-by-point response to Reviewer 2 (Anonymous).**

In this manuscript, Picton et al. use a range of remotely sensed datasets to examine recent glaciological changes within Vincennes Bay, East Antarctica. Amongst other interesting indicators of glacier change, they report, most notably, upon the rapid retreat of Vanderford Glacier's grounding line at a rate of 0.8 km/yr between 1996 and 2020.

Overall, the paper is generally well-presented, well-structured and is scientifically robust, and even considering the observations of Vanderford Glacier's behavior alone – which places it as the 4th fastest retreating glacier in Antarctica over the satellite era – I believe this manuscript will be of broad interest to the readership of The Cryosphere. For this reason, I recommend publication. Prior to publication, however, I believe the manuscript has several limitations in its current form which should be addressed. These limitations are detailed in my comments below.

We thank the anonymous reviewer for their positive feedback on the manuscript and constructive suggestions. Please find our responses to the general, minor and technical comments outlined below.

**4    General Comments**

*Vanderford grounding-line retreat.* While the retreat rate reported here is undoubtedly significant, the problem is that it is for the most part not a new finding. This is because, as the authors themselves allude to on Line 84, Rignot et al. (2019) have previously reported upon this behavior as observed between 1996 and 2017 (over which time they also find a retreat rate of 0.8 km/yr). In this regard there are two key issues with the manuscript in its current form:

1) The grounding-line retreat-related findings – as presently reported at least – are perhaps more incremental than the narrative of the manuscript would suggest.

2) Apart from the introduction (Line 84), no further acknowledgement of Rignot et al.'s earlier observations is included, which could be misconstrued by some as slightly disingenuous.

To remedy these issues, I would suggest the authors:

1) Rework the text to contextualize their findings more clearly alongside this earlier research. (In e.g. the abstract and conclusion, phrasing like "Our results confirm extensive grounding-line retreat..." is used, but this doesn't make it explicit that this is a confirmation of a previously documented observation). More explicit follow-up discussion of the fact that the author's observations show continued retreat since 2017 would also be beneficial, and serve to demonstrate that their research goes beyond that discussed by Rignot et al.

2) What's also new (and arguably much more interesting) relative to the simple trend of 0.8 km/yr reported here and in Rignot et al. (2019) is the seemingly step-wise, temporally variable patterns of retreat observed between 1996 and 2021 (Fig. 8). I think a more explicit/nuanced discussion of this phenomenon and its links with e.g. changes in bed topography/MISI (or otherwise) as seen in Fig. 9 would make for a much more interesting read, while again going beyond that described in Rignot et al. (2019). (See also my **Minor Comment** on Fig.9 below).

We appreciate that this could be misconstrued by some as slightly disingenuous, which was certainly not our intention. We have ensured to emphasise throughout the manuscript that our study supports a previous observation made by Rignot et al. (2019), with a similar rate of grounding line retreat being observed. Given the uncertainties associated with the ASAID and MOA datasets, the Bed-Machine bed elevation, as well as the difficulties in comparing the hinge-line and break-in-slope, we don't think a step-wise discussion is appropriate. However, we have altered the tone of the abstract, discussion and conclusion to emphasise the **persistent and sustained** nature of the grounding-line retreat across the study period.

*Structure/presentation of manuscript.* While generally well-presented/written overall, I believe the manuscript could also be overhauled in places (abstract, discussion and conclusion especially) to offer a more succinct / 'to the point' discussion of the key points and novel findings only. Perhaps most importantly, I think the structure of the discussion requires some careful refocusing, as at present

it contains a lot of unnecessary details which should either be moved to the Methods, Results or Supplementary Information. Elements of the Discussion and Conclusion also have the tendency to jump back and forth between ideas and/or from one sub-section to the next, which I think should be corrected for improved readability. (See my **Minor Comments** below for some examples of these sorts of issues).

We recognisde that parts of the manuscript, particularly the abstract, discussion and conclusion, could be made more succinct and corrected for improved readability. We have thus made a number of changes, with specific examples provided in our response to the minor comments outlined below.

**5    Minor Comments**

L13 - The ice surface velocity, thinning and GL position datasets are also derived from remotely sensed techniques, so I suggest rephrasing the sentence to better convey this point.

This sentence has been rephrased in order to better convey this point.

L13 - synthetic aperture radar.

This has been changed to state 'synthetic aperture radar'.

L34 - driven - dominated (I agree that these are the two main glaciers, but other parts of the coast are just as sensitive to oceanic influence and are now contributing to these trends too).

'driven' has been changed to 'dominated'.

L57 - 'measured to' - 'having undergone'?

'measured to' has been changed to 'having undergone'.

L60 - Weaker easterly winds relative to what and where? This was unclear to me. This line should be revised to clarify this.

This has been clarified by stating 'Wilkes Land is characterised by a 'warm shelf' regime, whereby weak easterly winds and an absence of dense water formation facilitates the intrusion of warm CDW onto the continental shelf (Thompson et al., 2018; Stokes et al., 2022).'

Figure 1 - Nice figure. Could 1b be underlain by hillshade and/or contours to make the topography 'pop out' more? In 1a, I also strongly recommend displaying the most recent GL (or ideally all of those included in your timeseries if the figure doesn't look too cluttered) to give the reader an instant sense of how each glacier has retreated through time. To help the reader find the locations referred to in the text (e.g., L81), please also add lat/lon graticules to both panels (and all other maps for that matter).

Figure 1b has been underlain by a hillshade in order to make the topography more clearly visible. Tick marks and longitude/latitude gridlines have also been added to all maps. However, the decision was made to only display the oldest GL position for two main reasons:

1) Figure 1 is primarily intended to introduce the regional context of the study area, particularly highlighting the location of Vincennes Bay relative to Totten Glacier and Law Dome, as well as displaying the positions of the central glacier flowlines and FT, GL and IN sampling boxes used for analysis. We believe that the inclusion of additional GL positions from the timeseries detracts from these important features, with Figure 1 appearing rather cluttered.

2) Showing the most recent GL position would effectively display the main results of the paper within the first figure, which we don't think would necessarily be appropriate. We would also like to emphasise that for Vanderford, Adams and Anzac, the most recent GL position would be the 2020 AIS CCI position, whilst for Bond East, Bond West and Underwood it would be the 2014 MOA position. Displaying these different datasets prior to their introduction, description and explanation in Section 2.5, may be rather confusing to the reader.

L101+ - Here, I suggest removing the methodological detail behind the analyses performed (terminus position, velocity, elevation, GL change) as this information is better placed in the following section.

Whilst we recognise that this paragraph includes some methodological detail, these details are summarised very succinctly. We think that the inclusion of this short paragraph provides an important link between the introduction and the methods section, outlining how the stated aim is to be achieved. We therefore suggest that this section improves the overall readability of the manuscript and should remain unchanged.

L109 - 'USGS Earth Explorer data repository, with...'

'data repository' has been added.

L112 - – Why was co-registration of the earlier scenes performed? I presume this pertains to the old ephemerides and DEMs used for geocoding Landsat 1-4 images, and thus the need to ensure spatially consistent imaging through time? Worth stating that here if so. Also, why was co-registration not performed for the ARGON mosaic? Considerable positional errors can exist in those images, and likely exist in the Kim et al. mosaic too.

The Landsat 1-4 image scenes were co-registered due to geolocation issues and the need to ensure spatially consistent imaging through time, as suggested above. This has been clarified within the manuscript. In contrast, co-registration was not conducted for the ARGON imagery as the mosaic produced by Kim et al. (2007) has already been orthorectified using GCPs. Nonetheless, the geolocation accuracy was manually checked before any subsequent analysis was conducted; the positions of coastal rock outcrops, nunataks and visibly stable ice features were observed to match those observed from the more recent Sentinel-2B scene. We have clarified that such prior co-registration and additional manual verification was conducted for the ARGON mosaic within Section 2.1.

L119 - ... are hereafter referred to as ...

Sentence structure has been changed to state 'are hereafter referred to as'.

L132 - ... the AutoRIFT feature tracking algorithm.

'auto-RIFT algorithms' has been changed to 'the AutoRIFT feature tracking algorithm'.

L135 - Typo? I believe ENVEO records date back to 2014?

This is correct and has been rephrased to provide clarification. Whilst the ENVEO velocity mosaics are available at a monthly resolution between 2014 and 2021, an additional sentence has been added to state that only those between 2019 and 2021 were used within this study.

L136 - Suggest rephrasing to say "... using a combination of coherent and incoherent feature tracking techniques". (following Nagler et al. (2021; doi:10.1109/igarss47720.2021.9553514) which should also be cited here).

Whilst the additional Nagler et al. (2021) citation has been included, this paper does not appear to explicitly mention 'coherent' or 'incoherent' techniques. We therefore decided not to include the suggested rephrasing.

L184-197 - As the authors are aware, the grounding line and hinge line are two different components of the grounding zone, the latter being a proxy for the former which cannot be detected from satellite imaging. This fact should be stated somewhere in this paragraph, if anything to make it clear that the two components haven't been conflated.

An additional sentence has been added, stating that the hinge-line is often used as a proxy for the actual grounding-line position.

L206/7 - '...was generated using...as applied to ERS-1 and ERS-2 imagery...'.

This sentence explains that the MEaSUREs grounding line product was not only generated using the same DInSAR technique as that used to generate the AIS CCI dataset, but that both datasets

also used ERS-1 and ERS-2 imagery collected in 1996. We therefore think that use of the word 'also' is preferable over use of the word 'as'. The sentence has thus been changed to read:

'The Making Earth Science Data Records for Use in Research Environments (MEaSUREs) grounding line product was generated using similar DInSAR techniques as previously described for the AIS CCI product, also applied to ERS-1 and ERS-2 imagery collected in 1996 (Rignot et al., 2016).'

L216 - Should read '...and most seaward, spatially continuous break-in-slope' (since multiple discontinuous breaks-in-slope can exist downstream over, for example, pinning points or ice rumples).

'spatially continuous break-in-slope' has been added.

L227 - Following my comments on L184-197, I think it's important to state here that it's difficult to compare break-in-slope and hinge line positions in a direct/like-for-like manner, because they're ultimately measuring two very different components of the grounding zone. Therefore, even given the standalone instrument/technique errors shown in Table 2, any changes in GL position identified from the two techniques should be interpreted with caution, with any further discussion restricted to those exhibiting pronounced retreat where we can be confident change represents a true signal.

This is a valid point and has been emphasised within Section 2.5. We state: 'The hinge-line and break-in-slope represent fundamentally different components of the grounding zone and therefore cannot be directly compared. Any observed grounding-line position change observed using the two different methods must thus be interpreted with caution, with further discussion restricted to pronounced signals and long-term trends that we can be confident are indicative of significant change.'

L230-234 - I'm not sure I follow this, and specifically why the two GLs would be so far apart. Geocoding issues? Tidal effects (if different imaging dates)? Both? Additional information explaining this would be good here.

We agree that this is slightly confusing, and would like to emphasise that we were also puzzled as to why the two grounding lines would be so far apart. This motivated our decision to measure retreat relative to the most landward observed 1996 position, in order to ensure that we reported the most conservative estimate of grounding line retreat at Underwood Glacier. No explicit explanation is provided in the documentation associated with the dataset (Rignot et al., 2016), with the 'Quality Assessment' simply stating:

"The standard error is ±100 m, with greater geolocation variations locally. In some cases, large (km) short-term and long-term migrations are present. The quality of the grounding line mapping depends on the satellite data used, the length of the interferometric baseline (short baselines yield more accurate positioning), the amplitude of the differential tides, phase coherence (high phase coherence means less noise), and the frequency of revisits."

Whilst tidal effects were considered, the metadata provided with each GL states that both positions were mapped using imagery from the same European Space Agency Earth Remote Sensing Satellite (ERS), across the same acquisition dates (1996/2/6, 1996/2/7, 1996/3/12 and 1996/3/13). We have added these additional details to the manuscript and emphasised that the reason for such geolocation differences remains unknown.

Fig.2 - Suggest noting the non-linear x-axis scale in the caption or, even better, editing the figures to show a linear scale.

The x-axis scale is linear, however the tick marks were placed at irregular spacings. This decision was made in an attempt to allow the reader to better understand the exact timings of the terminus digitisations, particularly across the earlier years of the record. For example, it can clearly be seen that the terminus positions of Vanderford, Adams and Anzac glacier were first observed in 1963, whereas Bond East, Bond West and Underwood Glacier were first observed in 1973. However, we appreciate that such irregular spacing may be confusing to the reader and potentially misinterpreted as a non-linear scale. The labelled tick marks have therefore instead been placed at regular 10-year intervals between 1960 and 2020, with minor tick marks shown every 5 years.

Fig.3 - Nice figure. Caption should read '...from the USGS Earth Explorer data repository' or

similar.

‘data repository’ has been added.

Fig.4 - Nice figure, but what does, for example ’Distance from VA’ mean on the x-axis? Does this literally mean from the ’VA’ label on Figure 1? If so, suggest changing the notion to read ’Distance downstream’ *or similar), and annotating VA to VA on both Fig.1 and the cross section profiles of Fig.4.

The Figure 4 x-axis caption has been changed to state ’Distance along VA flowline (km)’. The figure caption has also been edited to explicitly state that distance is measured in the along-flow direction, with 0 km representing the inland start point of each respective flowline.

L290 - Associated errors. This is great, but please also show these errors on Fig 5 too.

We recognise that the associated errors could be added to Figure 5, but because each individual graph displays the mean annual velocity across each of the IN, GL and FT boxes, this addition makes Figure 5 rather busy. Instead, we believe it is more helpful to maintain the clarity of Figure 5, but ensure that for each specific velocity value reported in Section 3.2, the associated error is stated. L308-310 [of the revised manuscript] has therefore been changed to state:

‘Whilst ice surface velocity was seen to increase by 12% across the FT of Anzac Glacier between 2009 and 2021 (Figure 5c), this velocity increase was not deemed notable, with the absolute value of acceleration (30 m/yr) being smaller than the associated error ($\pm$ 82 m/yr).’

Figs. 6 and 7 - These are clear, detail-rich figures. My one thought, however, is whether they could be merged somehow to save the reader flicking back and forth between figs?

The Smith et al. (2020) dataset shown in Figure 7 simply provides a value representing the average rate of surface elevation change observed between 2003 and 2019. In contrast, each of the Schröder et al. (2019) and Nilsson et al. (2022) datasets shown in Figure 6 represent monthly surface elevation change values. Whilst the Smith et al. (2020) dataset may be added to Figure 6 in the form of a linear trendline with a constant gradient, an arbitrary elevation would have to be chosen in 2003 which could potentially be misleading. Inclusion of such a linear trendline may also distract from the interannual variability captured by the 24-month rolling means calculated from the Schröder et al. (2019) and Nilsson et al. (2022) datasets. The decision was therefore made to not include the Smith et al. (2020) dataset within Figure 6. Instead, the Smith et al. (2020) dataset was primarily used to validate and provide further certainty to the long-term trends extracted from the Schröder et al. (2019) and Nilsson et al. (2022) datasets, with Figure 7 used to highlight the spatial variation in ice surface elevation change observed within Vincennes Bay.

L340 - Following my comment on L227, I think it’s important to mention here that the retreat observed falls greatly outside sensor error limits, and likely also any between-sensor uncertainties associated with combined DInSAR/break-in-slope comparisons.

This additional point has been added. We state ‘This magnitude of retreat falls greatly outside the estimated error values provided with each dataset (Table 2), and likely exceeds any uncertainties associated with comparing the hinge-line (AIS CCI and MEaSUREs) and break-in-slope (ASAID and MOA).’

Fig 8 - This is a nice figure, although I question whether it’s integral to the interpretation presented. I think a revised version of Fig. 9 with each GL position shown would be far more impactful, with Fig. 8 moved to the supplementary information instead. Either way, remove the word ’manually’ from the caption since the methods states that both break-in-slope and DInSAR mapping was carried out in this way.

We appreciate this comment, but think that Figure 8 is important and integral to the manuscript for two primary reasons:

1) As previously stated in the general comments, Figure 8 is useful for showing the temporally variable patterns of grounding line retreat observed between 1996 and 2020. It highlights that the grounding line positions mapped from optical imagery (ASAID and MOA datasets) were digitised

in near-identical locations at each of the Vincennes Bay outlet glaciers, with the notable exception of Vanderford Glacier. Such near-identical locations would not be visible to the reader if placed as vertical lines on Figure 9, which we think would detract from the associated discussion.

2) The symbology used within Figure 8 allows the reader to assess the different methods used to quantify grounding line retreat. We believe this is unique to the manuscript and provides important context for the later discussion of the importance of accurate DInSAR grounding line mapping.

Nonetheless, the figure caption has been changed to remove the word 'manually' as suggested.

Fig 9 - Please add each GL location to portray the temporal evolution of retreat more clearly, and if needed revise the results/discussion to reflect any temporal patterns of retreat this reveals (see also my **General Comment** on this point).

As discussed above, we don't think that Figure 9 is suitable for displaying the temporal evolution of grounding line retreat clearly. This is primarily due to the near-identical ASAID and MOA grounding line positions observed at 5 of the 6 studied glaciers, which would be difficult for the reader to observe as dated vertical lines. However, Figure 9 still provides important context of both the bed and ice surface elevation along each flowline. We therefore think there is value in keeping both Figure 8 and Figure 9 in the manuscript, and note that the total of 12 figures is not considered excessive for a paper published in *The Cryosphere*. As stated in our earlier response, we have altered the tone of the discussion to emphasise the **peristent and sustained** nature of the grounding-line retreat observed at Vanderford Glacier across the study period.

L368-375 - I have two comments about this paragraph. First, I believe this is a significant finding which could/should be better articulated both here and in the abstract and conclusion. By my understanding this places Vanderford Glacier as the 4th fastest retreating glacier in Antarctica (let alone East Antarctica) over the satellite era, only after Thwaites (0.8 kmyr = 3rd, as mentioned by the authors, Pine Island (0.9-1 kmyr= 2nd, Park et al., 2013; doi:10.1002/grl.50379) and Pope (3.3 kmyr = 1st, Millilo et al., 2022; doi:10.1038/s41561-021-00877-z).

We have emphasised the significance of the rapid rate of grounding line retreat observed at Vanderford Glacier more, both in this paragraph and within the abstract and conclusion. However, we have stated that Vanderford Glacier is the **third** fastest retreating glacier over the satellite era, **at the decadal scale**. This was motivated by the fact that the high rate of grounding-line retreat observed at Pope Glacier was over a significantly shorter time period. We believe this is an important distinction when making comparisons to the consistent and sustained decadal-scale trends observed at both Pine Island and Thwaites Glacier.

That said, I was a little surprised to see no discussion of Rignot et al. (2019) here, who following Line 84 report the same rate of retreat between 1996 and 2017 (i.e. overlapping 90% of the present study's observational period). As such, I believe this paragraph (and elsewhere) should be reworked to better contextualize the author's findings against this study.

[NB: I don't think the above suggestion will necessarily detract from the novelty of the present study, as long as it's made clear your observations show the *continued* rapid retreat of Vanderford Glacier to 2020].

As stated in our earlier response to the general comments, we have tried to emphasise throughout the manuscript that our study supports a previous observation made by Rignot et al. (2019), with a similar rate of grounding line retreat being observed.

L416-419 - So what caused this significant GL retreat then? Discussion of passive ice according to Furst et al. is good, but strikes me as a convenient diversion away from what should be the main focus of this paragraph. A revised discussion of the expected key drivers of this phenomenon will make for a much more compelling read, even if the interpretation is 'the causes remain unknown and an important region for future research' or the like.

We think that Section 4.1 outlines what may be causing the significant GL retreat, with clear discussion of the potential intrusion of warm mCDW along the Vanderford Trench at depth. However, we have tried to strengthen this discussion further by including a later suggestion made by the anonymous reviewer. We have emphasised that in Antarctica, the majority of basal melting is confined

to the GL, due to water column stratification and the role of pressure in determining the freezing point of water (Rignot et al., 2013). We discuss that such focussed melting may therefore explain the significant grounding-line retreat observed relative to the comparative lack of dynamic change.

Fig 11 - If GLs are added to Fig.1 (see my comments on that figure above), then I'd consider moving Fig.11 to the supplementary information to avoid figure repetition in the main text.

As explained previously, we think the addition of further GLs makes Figure 1 too cluttered and displays the main findings of the paper before the context has been established, potentially confusing the reader. We therefore think Figure 11 should remain in the manuscript and note that a total of 12 figures is not considered excessive for a paper published in *The Cryosphere*. It should also be noted that the Vanderford flowline is extended in Figure 11 and therefore serves an important role in highlighting the inland retrograde slope observed along the Vanderford Trench. We have, however, removed Figure 11a (ice surface velocity) in order to somewhat minimise the figure repetition.

L450-476 - How has landfast sea ice changed between 1996 and present? An increasing amount of research has shown the importance of landfast sea ice/mélange for controlling calving rates in Antarctica (by congealing together / buttressing ice fronts), which I think should also be noted here. (See papers by e.g. Greene et al. (doi:10.5194/tc-12-2869-2018), Francis et al. (doi:10.5194/tc-15-2147-2021), Fraser et al. (doi:10.5194/essd-12-2987-202), Arthur et al.(doi:10.1017/jog.2021.45), Christie et al. (doi:10.1038/s41561-022-00938-x) and Massom et al. (doi:10.1038/s41586-018-0212-1) for some recent examples of this in Antarctica).

L498-510 – OK, the majority of this paragraph goes on to discuss the importance of landfast sea ice for controlling terminus stability, but it seems largely out of place to me in this section. All things considered, to improve readability/clarity I would suggest merging Sections 4.2 and 4.3 into a new, single section which first overviews the observed glaciological changes (frontal retreat and thinning) and then discusses their possible links to sea-ice loss.

On a related note, recent work has also shown that decreased sea ice cover may also leave coastal regions vulnerable to the influence of storms or katabatic wind events which can disturb ocean surface slopes leading to enhanced strain-induced calving (see e.g. Francis et al. (doi:10.5194/tc-15-2147- 2021; 10.1029/2021JD036424) and Christie et al. (doi:10.1038/s41561-022-00938-x) for recently documented examples of this phenomenon).

To my mind, these sorts of atmosphere-sea ice interactions (sea-ice debuttressing, wind-induced strain) are more likely to explain the observed patterns of calving than the mechanism proposed by Miles et al. 2016 (at least over the relatively short timescales considered here), because in Antarctica the majority of basal melting is confined to the GL (e.g. Rignot et al., 2013; doi:10.1126/science.1235798). This phenomenon doesn't occur by coincidence, and is because mCDW resides at the same depth as the GL. Elsewhere (including at the relatively much more shallow ice-shelf fronts), mCDW does not physically interact with the ice in such a way that can drive rapid frontal fracture/calving, and thus the more moderate thinning signals observed at those locations instead reflect lagged responses resulting from historical perturbations at the GL.

In response to the above four comments, we have merged Section 4.2 and Section 4.3 into a single section entitled 'The role of sea-ice in controlling outlet glacier ice dynamics'. We have decided to keep the frontal retreat and thinning sections seperate, as they represent glaciological changes over very different temporal scales. However, we have strengthened this section by including additional discussion of the potential influence of decreased sea ice cover on the vulnerability of coastal regions to storm-generated swells. We have also altered the tone of the discussion to emphasise that sea-ice debuttressing has likely played a more significant role than the calving mechanism proposed by Miles et al. (2016), as suggested.

Section 4.4 – While interesting, most of this section reads like background information and less pertinent results which I'm not sure is best placed in (or integral to) the discussion. I suggest making this a supplementary discussion and alluding to it somewhere in Section 3.2.

We appreciate that this section is not integral to the main discussion of the paper. However, with Bond West Glacier observed to reach speeds in excess of Shirase Glacier, previously thought to be the fastest flowing outlet glacier in East Antarctica, we believe it will still be of interest to the readership

of *The Cryosphere.* We have therefore decided to retain this section within the manuscript.

Section 4.5 – Similar to Section 4.4, much of this section contains information which should belong in the Methods (especially given my comments on Lines 184-197 and 227). Following my suggestion for Line 340, I also think the discussion contained on Lines 539-545 could be removed as the signal-to-noise of break-in-slope-derived change relative to the 1996 DInSAR GL pick seems compelling. I further suspect that the pattern of retreat revealed by plotting each GL onto Fig.9 will support this conclusion.

We think that this section represents an important discussion point of the paper, summarising the main methodological finding of the study. This section therefore builds upon the findings of Figure 8, providing a unique assessment of the different methods used to quantify grounding line retreat. With this glaciological parameter representing such an important indicator of dynamic change, we believe this discussion will be of interest to the readership of *The Cryosphere.*

Section 5 – While acceptable as written, the conclusion is very long and could/should be overhauled to provide a much more succinct, punchy summary of the key findings and implications only. I expect this could be done in half a page or less. Structure-wise, I also find the discussion going from *GL retreat* to *terminus change* and then back to *GL retreat* to be confusing, so suggest that the ideas contained in the paragraphs beginning Lines 547 and 573 could be merged into one coherent narrative.

We don't consider three paragraphs to be too excessive for a conclusion, however we have made the conclusion more succinct wherever possible. We also appreciate that switching from *GL retreat* to *terminus change* and then back to *GL retreat* may be confusing to the reader and have therefore restructured the conclusion to form one coherent narrative.

L550 – Same comment as Lines 368-375.

As stated earlier, we have emphasised the significance of the rapid rate of grounding line retreat observed at Vanderford Glacier more, stating that it is the third-fastest retreating glacier in Antarctica, at the decadal scale.

L585 – In these days of reproducibility I would strongly encourage the authors to archive and openly share their terminus positions via a repository such as Cryoportal.

The manually digitised central flowlines, sampling boxes and terminus positions have been uploaded to the UK Polar Data Centre and can be accessed at: https://doi.org/10.5285/4D4BD383-F6BB-4476-9058-D883B7706B26. This dataset is under embargo until paper publication, but can be accessed by the reviewers if required.

Table S1 – Table Caption – For completeness, insert 'optical' after 'Details of'.

Caption has been changed to state 'Details of the optical satellite imagery used within this study'.

**6    Technical Comments**

L33 – WAIS in this instance is a pronoun and so should not be preceded by 'the'. Similarly,'embayment' should be capitalized. Multiple such blunders exist throughout the manuscript, so it will be worth carefully going through the text and weeding these out.

The sentence reads 'Recent mass loss from the WAIS has largely been concentrated...'. We therefore believe that in order to be grammatically correct, use of the word 'the' is required before 'WAIS'. We note that this convention is regularly adopted within the literature, including published papers within *The Cyrosphere.* Some recent examples include Maclennan et al. (2023) (https://doi.org/10.5194/tc-17-865-2023), Holland et al. (2022) (https://doi.org/10.5194/tc-16-5085-2022) and Schlemm et al. (2022) (https://doi.org/10.5194/tc-16-1979-2022).

However, we have ensured that 'embayment' is capitalised throughout the manuscript.

L37 – 'grounding line retreat' should read 'grounding-line retreat'. This should be changed throughout the manuscript, including in the title. Same for phrases such as 'ice-shelf thinning', 'ice-surface velocity', 'sea-ice production' etc.

'grounding line retreat' has been changed to 'grounding-line retreat' throughout the manuscript, including in the title. Such changes have also been completed for the suggested phrases outlined above.

L208 – Tense issue. Suggest rewording to: '...has an overall, associated error of +/- 100m'.

This has been reworded as suggested in order to correct the tense issue.

L221 – Grammar. Why does the use of MODIS imagery mean that the error is 250 m? Suggest rephrasing for improved clarity.

Further detail has been added in order to improve the clarity.

L291 – Typo? Think it should read '... at box IN accelerating from ...'.

This has been changed to state '... at box IN accelerating from ...', as suggested.

L323 – 'These trends'. What trends? Grammar. Perhaps the opening sentence of this paragraph could read something like: "Mirroring the elevation patterns observed by Schroder et al. (2019) and Nilsson et al. (2021), similar trends are found in the ICESat/ICESat-2-derived data of Smith et al. (2020) (Figure 7)".

Sentence has been changed to that suggested above.

L406 – Insert comma after '(Figure 6a)'.

Comma has been inserted after '(Figure 6a)'.

L560 – Missing citation?

This sentence has been removed.

Section 7 – Text missing?

A link to the supplementary material associated with this paper will be provided upon publication